# Assessing the role of children in the COVID-19 pandemic in Belgium using perturbation analysis

Leonardo Angeli [1] ✉, Constantino Pereira Caetano[2], Nicolas Franco[1,3], Pietro Coletti [1,4], Christel Faes[1], Geert Molenberghs[1,5], Philippe Beutels [6], Steven Abrams [1,7], Lander Willem [6,7] & Niel Hens [1,6]

Understanding the evolving role of different age groups in virus transmission is essential for effective pandemic management. We investigated SARS-CoV-2 transmission in Belgium from November 2020 to February 2022, focusing on age-specific patterns. Using a next generation matrix approach integrating social contact data and simulating population susceptibility evolution, we performed a longitudinal perturbation analysis of the effective reproduction number to unravel age-specific transmission dynamics. From November to December 2020, adults in the [18, 60) age group were the main transmission drivers, while children contributed marginally. This pattern shifted between January and March 2021, when in-person education resumed, and the Alpha variant emerged: children aged under 12 years old were crucial in transmission. Stringent social distancing measures in March 2021 helped diminish the noticeable contribution of the [18, 30) age group. By June 2021, as the Delta variant became the predominant strain, adults aged [18, 40) years emerged as main contributors to transmission, with a resurgence in children's contribution during September-October 2021. This study highlights the effectiveness of our methodology in identifying age-specific transmission patterns.

The COVID-19 pandemic has led to an unprecedented global health crisis, demanding a detailed analysis of the factors driving its transmission dynamics. As such, social contact patterns, immune response, and infectiousness showed strong age-related heterogeneity early on in different contexts[1–3]. These age-related differences have a critical impact on transmission and key epidemiological outcomes such as disease incidence and prevalence, hospitalisation, and mortality[4–6]. Understanding the extent to which these differences influence such vital outcomes is essential for the complex task of developing comprehensive policies that address the evolving challenges posed by a global pandemic like COVID-19. In Belgium, multiple studies[7–12] have

underscored the importance of analysing age-specific transmission patterns to inform effective public health interventions. Our study aligns with this objective by offering analytical tools that aim to clarify the role of age in transmission within the complex landscape of evolving Non-Pharmaceutical Interventions (NPIs), vaccination efforts, and the emergence of virus variants in Belgium. These insights can aid policymakers in crafting balanced strategies that consider not only virus containment but also broader societal implications, such as maintaining economic stability[13,14] and safeguarding mental health[15,16]. In particular, considering younger populations, it is crucial to weigh the trade-offs of stringent measures such as extensive school closures

[1]Data Science Institute, I-BioStat, Hasselt University, Hasselt, Belgium. [2]Center for Computational and Stochastic Mathematics, Instituto Superior Técnico, University of Lisbon, Lisbon, Portugal. [3]Namur Institute for Complex Systems (naXys) and Department of Mathematics, University of Namur, Namur, Belgium. [4]Institute of Health and Society (IRSS), UCLouvain (Université catholique de Louvain), Brussels, Belgium. [5]L-BioStat, KU Leuven, Leuven, Belgium. [6]Centre for Health Economics Research and Modelling Infectious Diseases, Vaccine & Infectious Disease Institute, University of Antwerp, Antwerp, Belgium. [7]Department of Family Medicine and Population Health (FAMPOP), University of Antwerp, Antwerp, Belgium. ✉e-mail: leonardo.angeli@uhasselt.be

and lockdowns. While these interventions have proven effective in reducing transmission and disease prevalence[7,9,11,17], research also highlights their negative impact on children's mental health and educational outcomes, particularly for those from socio-economically vulnerable backgrounds[18–21]. The resulting stress on parents further emphasises the need for robust support systems during these times[18,22]. We will dwell on the children's roles in transmission, offering insights to help balance these trade-offs.

We use data from the CoMix social contact survey[7,23,24], a longitudinal, multi-country survey capturing the social interactions of representative samples of individuals including a broad range of demographic variables such as age, gender, region of residence. Specifically, we use Belgian data to inform age-specific social contact matrices over 34 consecutive survey waves from November 2020 to February 2022.

Our approach integrates the evolution of population susceptibility, the emergence of virus variants of concern (VOCs), vaccination rollout, and the impact of NPIs into the analysis. By longitudinally evaluating and interpreting *sensitivity indices* introduced in a related study[25], we aim to clarify the distinct roles of various age groups in virus spread, with particular emphasis on the evolving role of paediatric populations. This study is grounded in an age-structured Susceptible, Exposed, Infectious, and Recovered (SEIR) model[8,11], from which we derive our key analytical tool, the Next Generation Matrix (NGM or $\mathbf{K}$)[26,27].

The NGM models how an infected individual from a specific age group contributes to new infections in the same or different age groups over successive generations of disease spread[26]. Each element $k_{ij}$ of the matrix quantifies the expected number of secondary cases in age group $j$ produced by an infectious individual in age group $i$. The spectral radius of this matrix, $\rho(\mathbf{K})$, corresponds to the basic reproduction number, $R_0$, which is crucial for understanding the potential for an outbreak in a fully susceptible population. By updating the NGM to account for changes in susceptibility over time, our study provides insights into the variation of the effective reproduction number ($R_t$) throughout the study period. The said sensitivity indices are derived from the sensitivity analysis of $R_t$ as a function of the NGM entries. In this context, *sensitivity analysis* refers to using differential calculus to explore how changes in specific parameters influence a function of those parameters. However, the term is used to describe a variety of exercises, with a common example being the testing of findings robustness against changes in model assumptions. To avoid any confusion, we retain the term *sensitivity* for the indices but refer to the process as *perturbation analysis* throughout the study for clarity.

Unravelling age-specific transmission patterns has posed considerable challenges throughout the pandemic, particularly when it comes to understanding the role of children. Early studies suggested that children played a minimal role in transmission compared to adults[2,28]. However, this may have been underestimated due to the higher incidence of asymptomatic or mild cases in children and their reduced exposure during school closures[3,6]. Later research highlighted the noteworthy role of children, particularly in school and household settings, in virus transmission[11,29–32]. The spread of more transmissible variants, such as Alpha[33–38], Delta[39–43], and Omicron[33,44,45] further increased the prevalence and transmission risk within the paediatric population globally[32,46]. Notably, this became more pronounced during the phased reopening of schools, with studies revealing a strong correlation between school operations and broader transmission dynamics[31,47].

Our results build on these findings, highlighting children as key contributors to transmission during school reopenings and the relaxation of NPIs. In addition, young adults under 40 consistently played a substantial role, emerging as the main contributors to transmission when averaged over the entire study period. This study provides a nuanced understanding of how different age groups have contributed to SARS-CoV-2 transmission in Belgium over time.

In the following Results section, we highlight key shifts in transmission dynamics resulting from the interaction of behavioural factors (evolving NPIs and contact patterns) and epidemiological factors (vaccination and VOCs circulation). We stress that our results are inherently sensitive to the availability and quality of the data and parameter estimates that inform the model. In addition to the uncertainty inherent in the social contact data, a limitation of our methodology is the lack of serial serological survey data, would have enhanced the precision of age-specific susceptibility estimates over the observation period. To address this gap, we employed an extended stochastic compartmental model[48], calibrated on early pandemic serological data, COVID-19-related hospitalisations and deaths, to simulate susceptibility across age groups. Further methodological details are available in the Methods Section.

## Results

Our analysis explores the role of different age groups in SARS-CoV-2 transmission in Belgium from November 2020 to February 2022. The population is categorised into nine age groups aligned with the Belgian school system. Our observation begins with wave 9 of the CoMix survey, as earlier waves did not include data on contacts of individuals under 18 years of age[7], a key focus of our study. Data from survey wave 9, collected from November 11 to 18, 2020, reflect the Belgian population during the second lockdown, which was implemented on November 2, 2020, following a peak in infections, hospitalisations, and deaths around November 4[49].

We present the results of our analysis on age-specific transmission dynamics using key indices derived from the NGM, including Cumulative Sensitivity ($\bar{s}_{j,t}$), Infective Value ($v_{j,t}$), and Cumulative Elasticity ($\bar{e}_{j,t}$). These indices, summarised in Table 1, provide insights into how each age group influences $R_t$ and overall transmission. A composite index, $S_{j,t}$, integrates these measures to classify age groups according to their contribution to transmission over the study period. The subscript $j$ indicates the age group and the superscript $t$ indicates the specific point in time corresponding to each of the consecutive CoMix survey waves covering the study period (see Table 2).

For a better comparison, age groups are displayed in four quadrants in Figs. 1, 2, and 3, based on their cumulative elasticity and

**Table 1 | Interpretation of high values for the indices used to define the role of different age groups in SARS-CoV-2 transmission**

| Index | Interpretation |
|---|---|
| Cumulative Sensitivity ($\bar{s}_{j,t}$) | High values indicate expected high incidence in this group, at time $t$; $R_t$ is highly sensitive to changes in infectiousness and contacts within this group. |
| Infective Value ($v_{j,t}$) | High values indicate a high force of infection acting on this group, at time $t$; $R_t$ is highly sensitive to changes in susceptibility and contacts within this group. |
| Cumulative Elasticity ($\bar{e}_{j,t}$) | High values indicate a high proportional contribution to $R_t$, at time $t$, making it highly sensitive to changes in this group's parameters. |
| Composite Index ($S_{j,t}$) | Summarises the role of an age group in virus spread at time $t$; it ranges from 0 (all indices of group $j$ are below average) to 3 (all indices are above average). |

## Table 2 | Evolution of key quantities over time

| Age | wave 9 | wave 10 | wave 11 | wave 12 | wave 13 | wave 14 | wave 15 | wave 16 | wave 17 | wave 18 | wave 19 | wave 20 |
|---|---|---|---|---|---|---|---|---|---|---|---|---|
| [0,6) | 0 | 0 | 0 | 0 | 2 | 3 | 2 | 2 | 3 | 1 | 0 | 0 |
| [6,12) | 0 | 0 | 0 | 0 | 2 | 2 | 2 | 2 | 3 | 1 | 0 | 1 |
| [12,18) | 0 | 0 | 0 | 0 | 1 | 1 | 1 | 0 | 0 | 1 | 0 | 0 |
| [18,30) | 3 | 3 | 3 | 3 | 3 | 2 | 3 | 2 | 2 | 3 | 3 | 3 |
| [30,40) | 3 | 3 | 3 | 2 | 1 | 1 | 1 | 2 | 1 | 2 | 3 | 3 |
| [40,50) | 3 | 3 | 3 | 3 | 1 | 1 | 2 | 1 | 0 | 3 | 3 | 2 |
| [50,60) | 1 | 3 | 2 | 3 | 1 | 0 | 2 | 3 | 0 | 2 | 3 | 3 |
| [60,70) | 0 | 0 | 0 | 0 | 0 | 0 | 0 | 0 | 0 | 0 | 1 | 1 |
| 70+ | 0 | 0 | 0 | 3 | 0 | 0 | 0 | 0 | 0 | 0 | 0 | 0 |
| Stringency index | 65.74 | 63.89 | 60.19 | 60.19 | 60.19 | 60.19 | 62.96 | 62.96 | 62.96 | 62.96 | 70.37 | 75.93 |
| School & Work (%) | 83 | 67 | 50 | 50 | 50 | 50 | 50 | 50 | 50 | 50 | 83 | 83 |
| Travel ban & pub. transport (%) | 60 | 60 | 60 | 60 | 60 | 60 | 80 | 80 | 80 | 80 | 80 | 100 |
| Avg. contacts/day | 4.20 | 4.70 | 4.74 | 4.31 | 5.99 | 5.67 | 6.24 | 4.48 | 6.25 | 5.21 | 4.18 | 4.21 |
| Avg. contacts/day (-18) | 2.86 | 4.25 | 3.46 | 5.08 | 11.88 | 11.67 | 11.37 | 7.60 | 12.75 | 8.59 | 4.47 | 4.90 |
| Avg. contacts/day (+18) | 4.54 | 4.82 | 5.07 | 4.11 | 4.48 | 4.14 | 4.93 | 3.68 | 4.59 | 4.34 | 4.11 | 4.03 |
| $R_t$ (mean) | 0.90 | 0.76 | 0.77 | 0.88 | 1.02 | 1.09 | 1.07 | 1.04 | 1.04 | 1.07 | 1.08 | 1.01 |
| Date | 13/Nov/20 | 30/Nov/20 | 11/Dec/20 | 24/Dec/20 | 06/Jan/21 | 20/Jan/21 | 05/Feb/21 | 20/Feb/21 | 02/Mar/21 | 18/Mar/21 | 02/Apr/21 | 16/Apr/21 |
| **Age** | **Wave 21** | **Wave 22** | **Wave 23** | **Wave 24** | **Wave 25** | **Wave 26** | **Wave 27** | **Wave 28** | **Wave 29** | **Wave 30** | **Wave 31** | **Wave 32** |
| [0,6) | 3 | 1 | 2 | 2 | 0 | 2 | 1 | 0 | 0 | 2 | 3 | 2 |
| [6,12) | 3 | 2 | 3 | 2 | 2 | 1 | 3 | 0 | 0 | 2 | 3 | 1 |
| [12,18) | 1 | 1 | 2 | 2 | 2 | 1 | 0 | 0 | 0 | 1 | 0 | 1 |
| [18,30) | 1 | 3 | 1 | 2 | 1 | 3 | 3 | 3 | 3 | 1 | 0 | 3 |
| [30,40) | 1 | 3 | 2 | 2 | 3 | 3 | 3 | 3 | 3 | 1 | 2 | 3 |
| [40,50) | 0 | 2 | 2 | 2 | 0 | 2 | 0 | 3 | 0 | 3 | 0 | 3 |
| [50,60) | 0 | 1 | 0 | 0 | 0 | 0 | 0 | 2 | 2 | 1 | 0 | 1 |
| [60,70) | 0 | 0 | 0 | 0 | 0 | 0 | 0 | 2 | 0 | 1 | 0 | 0 |
| 70+ | 0 | 0 | 0 | 0 | 0 | 0 | 0 | 2 | 0 | 0 | 0 | 0 |
| Stringency index | 60.19 | 50.93 | 50.93 | 50.93 | 50.93 | 50.93 | 50.93 | 47.22 | 47.22 | 43.06 | 43.06 | 32.18 |
| School & Work | 50 | 50 | 50 | 50 | 50 | 50 | 50 | 50 | 50 | 50 | 50 | 33.33 |
| Travel ban | 60 | 60 | 60 | 60 | 60 | 60 | 60 | 60 | 60 | 60 | 60 | 60 |
| Avg. contacts/day | 5.43 | 5.78 | 5.26 | 5.55 | 6.27 | 5.23 | 5.20 | 5.47 | 5.11 | 5.61 | 6.84 | 6.75 |
| Avg. contacts/day (-18) | 11.27 | 8.79 | 9.68 | 8.90 | 9.65 | 5.99 | 5.92 | 4.73 | 4.55 | 8.53 | 12.09 | 9.97 |
| Avg. contacts/day(+18) | 3.94 | 5.01 | 4.13 | 4.70 | 5.40 | 5.04 | 5.01 | 5.66 | 5.25 | 4.86 | 5.50 | 5.93 |
| Rt (mean) | 0.90 | 0.79 | 0.80 | 0.92 | 1.02 | 1.10 | 1.10 | 1.07 | 1.05 | 1.04 | 1.05 | 1.09 |
| Date | 28/Apr/21 | 16/May/21 | 28/May/21 | 12/Jun/21 | 23/Jun/21 | 10/Jul/21 | 24/Jul/21 | 06/Aug/21 | 20/Aug/21 | 03/Sep/21 | 17/Sep/21 | 01/Oct/21 |
| **Age** | **Wave 33** | **Wave 34** | **Wave 35** | **Wave 36** | **Wave 37** | **Wave 38** | **Wave 39** | **Wave 40** | **Wave 41** | **Wave 42** | | |
| [0,6) | 3 | 3 | 0 | 1 | 1 | 0 | 1 | 0 | 3 | 1 | | |
| [6,12) | 2 | 3 | 0 | 2 | 1 | 0 | 0 | 0 | 2 | 1 | | |
| [12,18) | 1 | 1 | 1 | 2 | 1 | 1 | 1 | 1 | 1 | 1 | | |
| [18,30) | 2 | 1 | 3 | 2 | 2 | 3 | 3 | 1 | 3 | 3 | | |
| [30,40) | 3 | 3 | 3 | 3 | 3 | 3 | 3 | 3 | 3 | 3 | | |
| [40,50) | 3 | 2 | 2 | 2 | 3 | 3 | 3 | 2 | 1 | 3 | | |
| [50,60) | 0 | 0 | 2 | 1 | 1 | 3 | 3 | 0 | 1 | 1 | | |
| [60,70) | 0 | 0 | 1 | 0 | 0 | 0 | 0 | 0 | 0 | 0 | | |
| 70+ | 0 | 0 | 0 | 0 | 0 | 0 | 0 | 1 | 0 | 1 | | |
| Stringency index | 32.10 | 32 | 31.93 | 36.83 | 36.75 | 33.90 | 33.87 | 30.13 | 30.08 | 29.85 | | |
| School & Work | 33.33 | 33.33 | 33.33 | 33.33 | 33.33 | 33.33 | 33.33 | 16.67 | 16.67 | 16.67 | | |
| Travel ban | 60 | 60 | 60 | 60 | 60 | 40 | 40 | 40 | 40 | 40 | | |
| Avg. contacts/day | 6.38 | 6.06 | 5.32 | 4.75 | 4.68 | 4.33 | 4.06 | 4.88 | 4.31 | 5.23 | | |
| Avg. contacts/day (-18) | 10.21 | 10.72 | 6.35 | 7.93 | 6.32 | 4.23 | 4.54 | 5.75 | 6.95 | 7.90 | | |
| Avg. contacts/day(+18) | 5.41 | 4.87 | 5.06 | 3.94 | 4.26 | 4.36 | 3.93 | 4.66 | 3.63 | 4.55 | | |
| Rt (mean) | 1.15 | 1.21 | 1.21 | 1.17 | 1.11 | 1.05 | 1 | 0.96 | 0.94 | 0.94 | | |
| Date | 14/Oct/21 | 29/Oct/21 | 12/Nov/21 | 25/Nov/21 | 10/Dec/21 | 25/Dec/21 | 07/Jan/22 | 21/Jan/22 | 03/Feb/22 | 19/Feb/22 | | |

The table presents the progression of the $S_{i,t}$ index across different CoMix survey waves for various age groups, listed in the first 9 rows of each panel. It also includes the stringency index[50] of NPIs, detailing specific measures affecting schools, workplaces, travel, and public transport. Average daily contact rates for the overall population, split by adults and minors, are provided, along with the mean $R_t$ values for each wave, estimated from positive PCR tests[78]. This comprehensive data offers insights into the evolving dynamics of SARS-CoV-2 transmission across demographic segments over time.

sensitivity values at specific points in time. Each quadrant corresponds to an epidemiological category that defines the age-specific role in transmission dynamics. The main *Contributors* are groups with above-average cumulative sensitivity and elasticity, indicating the strongest influence on $R_t$ and key targets for control measures; *Effective Spreaders* have above-average cumulative sensitivity but below-average cumulative elasticity; *Incidental Spreaders* have above-average elasticity but below-average cumulative sensitivity; and *Marginal Contributors* have below-average values for both indices, playing a minimal role in transmission. The division of quadrants is based on average cumulative sensitivity and elasticity values, defined as $\tilde{s}_{avg} = 1$ and $\tilde{e}_{avg} = 0.11$. Further explanation on the interpretation and derivation of these indices can be found in the Methods Section and in the Supplementary Methods.

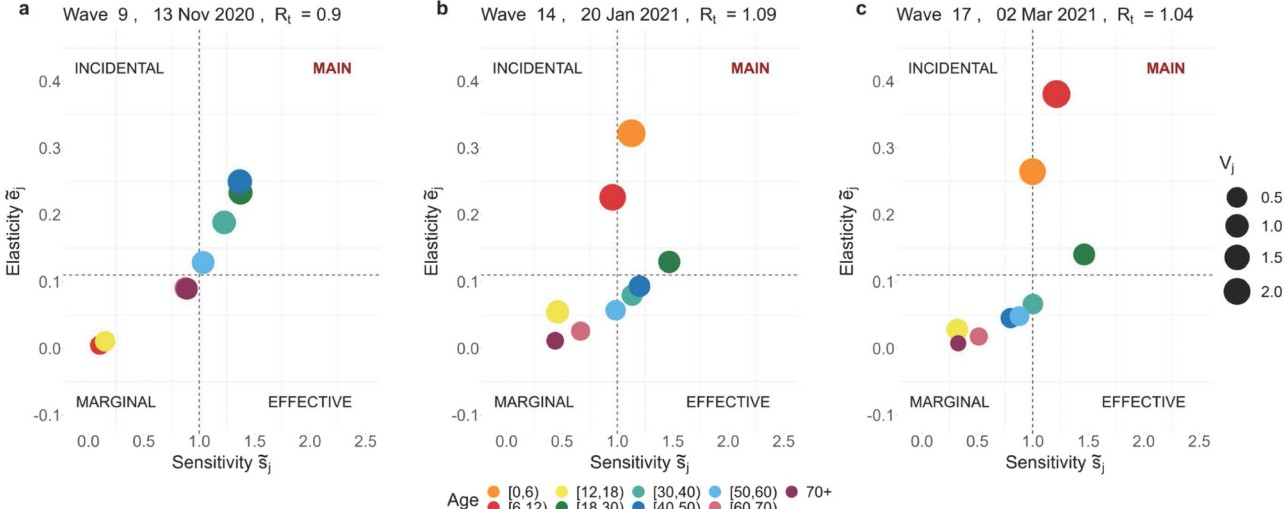

**Fig. 1 | Return to school in early 2021.** Age-specific cumulative sensitivity ($\bar{s}_{j,t}$) and elasticity ($\bar{e}_{j,t}$) values are visualised at times $t$ = 13, 14, 17. These correspond to different phases: (**a**) before-school reopenings, (**b**) after-school reopenings, and (**c**) before the Easter break. The quadrants---Main, Incidental, Effective, and Marginal--- reflect the various roles in virus transmission based on cumulative sensitivity and elasticity values. The size of the dots represents the infective value $v_{j,t}$, which averages 0.92 over the observation period.

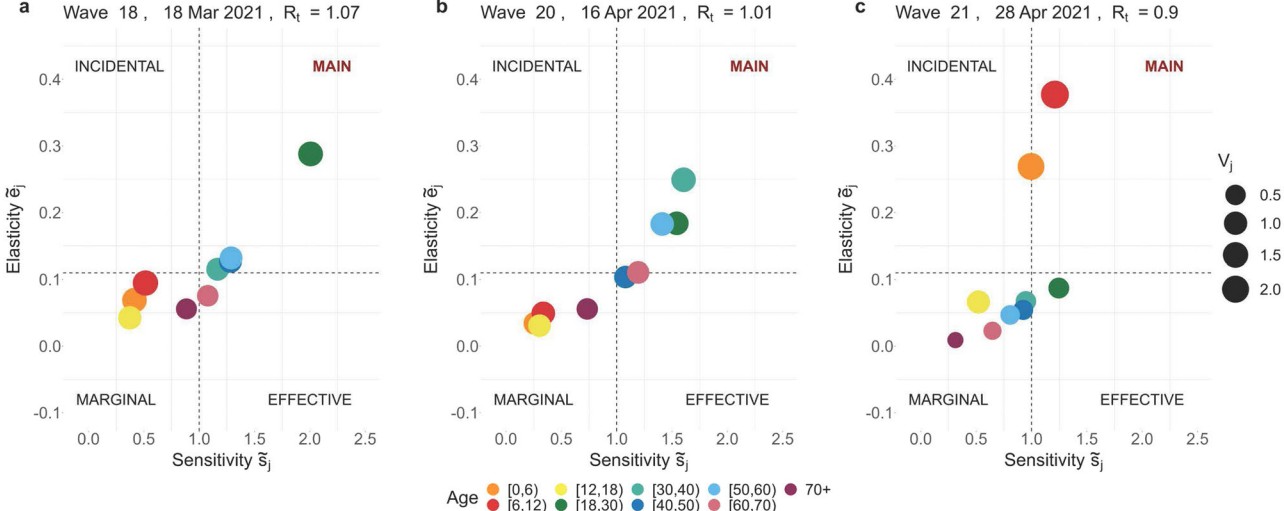

**Fig. 2 | Easter break 2021.** Age-specific cumulative sensitivity ($\bar{s}_{j,t}$) and elasticity ($\bar{e}_{j,t}$) values are visualised at times $t$ = 18, 20, 21. These correspond to different phases: (**a**) before the Easter Pause, (**b**) during the Easter Pause, (**c**) upon control measures relaxation. The quadrants---Main, Incidental, Effective, and Marginal--- reflect the various roles in virus transmission based on cumulative sensitivity and elasticity values. The size of the dots represents the infective value $v_{j,t}$, which averages 0.92 over the observation period.

## 13/11/20-24/12/20, survey waves 9 to 12

During this period, the stringency of the NPIs in place declined, with the stringency index[50] dropping from 65.74 in early November 2020 to 60.19 in late December 2020 (Table 2). This variation corresponds to the return to in-person education for primary and secondary schools from November 13, 2020, and the subsequent reopening of non-essential shops from December 1, 2020[51]. This period saw $R_t$ consistently below 1, indicating a controlled virus spread. Despite the easing of restrictions, the population-weighted average of daily reported contacts remained relatively stable, as well as the age-specific proportion of susceptible individuals (see Table 2 and Fig. 4). In this context, adults aged 18–60 years emerged as the main contributors to SARS-CoV-2 transmission. The index $S_{j,t}$ peaked at value 3 for these age groups, due to above-average indices $\bar{e}_{j,t}$, $\bar{s}_{j,t}$, and $v_{j,t}$ (see Fig. 1a).

## 06/01/21-02/03/21, survey waves 13 to 17

During this time interval, the NPIs remained moderately stringent, with an average stringency index of 61.9 (Table 2). This phase was marked by the continuation of in-person education, starting January 6, 2021, in the backdrop of increasing circulation of the Alpha VOC. Notably, the $R_t$ jumped from 0.88 (December 11, 2020) to 1.02 (January 6, 2021). The population-weighted average of daily contacts rose to a local maximum of 6.25 in wave 17 (March 2, 2021), when individuals under 18 years reported an average of 12.75 contacts per day (see Table 2). In the Supplementary Discussion, we elaborate on how this sharp change in the contact network supports the emergence of children aged 0 to 12 years as incidental virus spreaders. The cumulative elasticity ($\bar{e}_{j,t}$) and infective value ($v_{j,t}$) indices remained above average throughout this period, as reflected in an $S_{j,t}$ consistently equal to 2 or higher. Children's role in virus transmission was notable in January and March

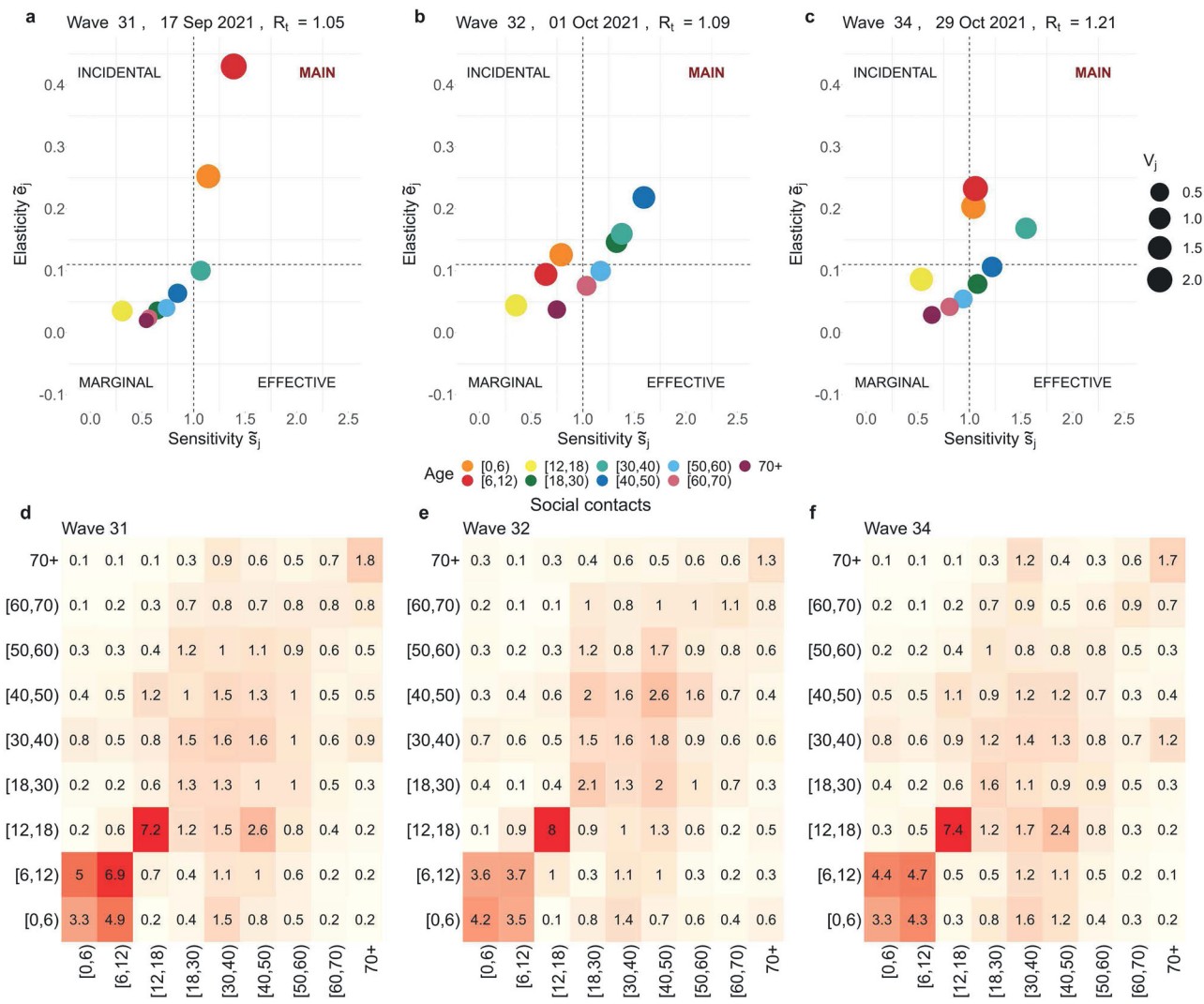

**Fig. 3 | Return to school in September 2021.** Age-specific cumulative sensitivity ($\bar{s}_{j,t}$) and elasticity ($\bar{e}_{j,t}$) values are visualised at times $t = 31, 32, 34$. These correspond to different phases: (**a**) return to school in September 2021, (**b**) re-opening pubs and dancing clubs, (**c**) new wave of contagions in autumn 2021, with a moderate increase in hospitalisations. The quadrants---Main, Incidental, Effective, and Marginal---reflect the various roles in virus transmission based on cumulative sensitivity and elasticity values. The size of the dots represents the infective value $v_{j,t}$, which averages 0.92 over the observation period. Panels (**d**–**f**) display the social contact matrices for the same survey waves. We underline the marginal role of the $[0, 12)$ group, with their high number of contacts of an extremely assortative nature.

2021, when $S_{j,t} = 3$ for $t = 14$ (January 20, 2021) and $t = 17$ (March 2, 2021). Specifically, the cumulative elasticity peaks are observed at $\bar{e}_{[0,6),14} = 0.32$ and $\bar{e}_{[6,12),17} = 0.38$, as shown in Fig. 1b and c. Children in the age groups $[0, 6)$ and $[6, 12)$ are classified as main contributors, respectively. These findings align with a generalised increase in new cases observed in Belgium from late February onwards[52], particularly in the 6–11 age band (school grades 1–6)[46], culminating in a peak at the end of March.

### 18/03/21-28/04/21, survey waves 18 to 21
Following the surge in cases and hospitalisations, which peaked in late March 2021 and largely originated from school and work settings[46,51,52], Belgium implemented a set of stricter NPIs, referred to as the "Easter Pause"[51]. During this period, from March 25 to April 19, 2021, the stringency index rose sharply to 75.93 on April 16, 2021 (Table 2). The perturbation analysis identifies adults in the age group $[18, 30)$ as main contributors to transmission, as shown in Fig. 2a. On March 18, 2021, the group exhibited an $S_{[18,30),18} = 3$, reflecting a high proportional contribution to an $R_t$ of 1.07 ($\bar{e}_{[18,30),18} = 0.29$) and a notably high cumulative sensitivity index ($\bar{s}_{[18,30),18} = 1.95$).

Throughout the Easter Pause (waves 19 to 20), we observe an increase in all the indices for adults aged 18 to 60 years, while $R_t$ remains above 1 (see Table 2 and Fig. 2b). The easing of restrictions (April 19, 2021), marked by a drop in the stringency index to 60.19, was driven by a reduction in $R_t$ below 1, a decline in daily new SARS-CoV-2 cases, and increased vaccination uptake[49,51,52]. Following the relaxation of NPIs, the population-weighted average number of contacts rose from 4.21 (wave 20) to 5.43 (wave 21), with a notable increase in social interactions among individuals under 18 years, who reported an average of 11.27 daily contacts (Table 2). On April 28, 2021, the perturbation analysis revealed a marked contrast between young adults ($[18, 30)$ age group) and younger individuals ($[0, 12)$ age group), with a notable decline in elasticity for the former and a steep rise in cumulative sensitivity and elasticity for the latter, as shown in Figs. 2c and 5. This trend is reflected in the values $\bar{e}_{[18,30),21} = 0.09$, $\bar{e}_{[0,6),21} = 0.27$, and $\bar{e}_{[6,12),21} = 0.38$, representing the proportional contributions to an $R_t$ of 0.9. At the same time, above-average cumulative sensitivity indices, allowed us to classify youngsters aged 0 to 12 years as the main contributors to transmission. Concurrently, Fig. 4 illustrates a noticeable drop in the age-specific proportion of susceptible

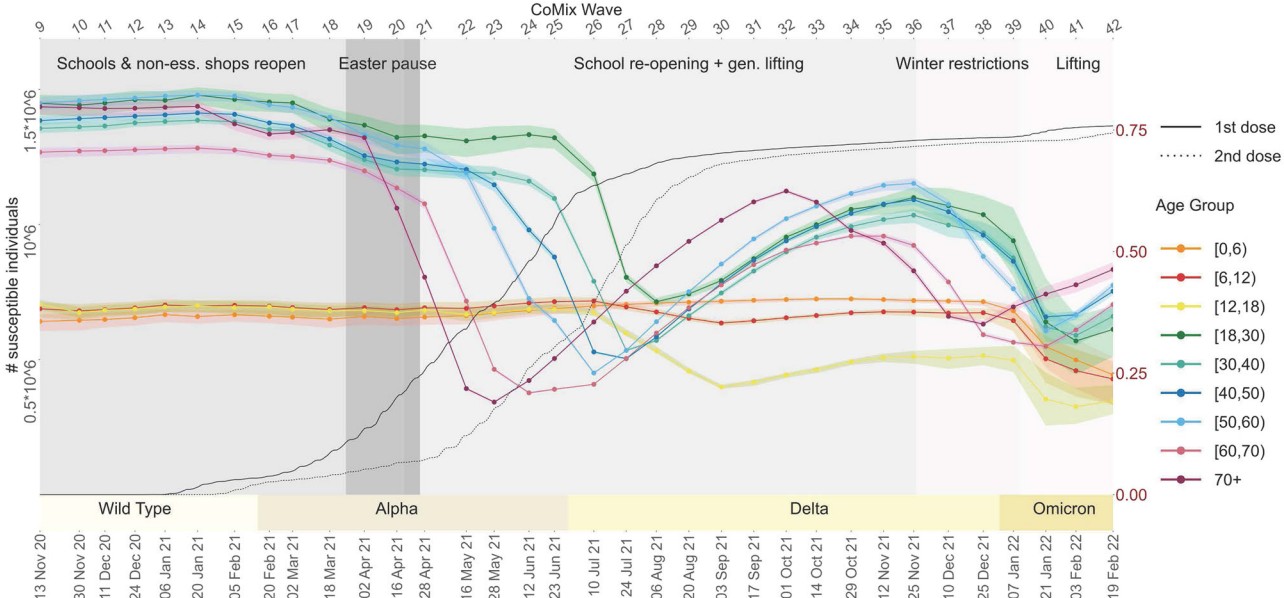

**Fig. 4 | Evolution of the susceptible compartment over time.** The curves depict the estimated daily number of individuals in the susceptible compartment by age group across the study period. This compartment includes individuals naive to both infection and vaccination, as well as those in the four compartments of waning immunity (see Methods Section for details). The points represent the mean estimates of the age-specific number of susceptibles at each point in the study period, with shaded areas showing 95% confidence intervals.

individuals aged 60 and above (wave 21). This trend corresponds to a reduction in the proportional contribution of individuals over 60 to transmission (see Fig. 5). A deeper exploration of the interplay between these factors can be found in the Supplementary Discussion (see Supplementary Fig. 5).

### 16/05/21- 20/08/21, survey waves 22 to 29

From May to August 2021, the gradual relaxation of NPIs continues in Belgium, as indicated by a further drop in the stringency index (50.9 in May and 47.2 in August 2021) - see Table 2 and Fig. 6. This phase initially corresponded with a reduction in transmission levels marked by an $R_t = 0.79$ on May 16, 2021(wave 22). However, the $R_t$ increased consistently and remained above 1 starting June 23, 2021 (wave 25). At this time in Belgium, the Delta VOC replaced the Alpha VOC as the predominant circulating SARS-CoV-2 strain. Characterised by increased transmissibility and potential to evade vaccine-induced immunity[39,41,42], the Delta VOC represented over 50% of the sequenced new cases at the beginning of July 2021[53,54]. Concurrently, the progress of vaccination across adult age groups, particularly the accelerated rollout for young adults ([18, 30)) from early summer 2021, and its subsequent extension to individuals aged [12, 18) after the EMA approved the Pfizer/BioNTech vaccine for those over 12 years old[49,51,52], altered the population's susceptibility profile. Figure 4 shows a steep decline in susceptibility within the [18, 40) age group during June and July 2021 (waves 25 to 27) and within the [12, 18) age group between July and August (waves 26 to 29). By late August 2021, a gradual rebound in susceptibility is observed across all adult groups, likely due to waning vaccine immunity and increased susceptibility to the Delta VOC[39-41]. During May-June 2021 (waves 22 to 25), individuals aged [12, 18) exhibited high elasticity and an increasing contribution to transmission, in contrast to their marginal role until late April (wave 21), as shown in Fig. 5. While this trend coincides with an increase in observed daily contacts (Supplementary Fig. 13), this alone does not fully explain the shift in age-specific contribution patterns reflected in cumulative elasticity indices. Elasticity patterns at waves 14 (Fig. 1b) and 31 (Fig. 3a) provide a counterexample, demonstrating that the [12, 18) age group can still play a marginal role in transmission despite a

high number of contacts. Further analysis of the cumulative elasticity gradient with respect to the NGM's columns suggests that the shift in social mixing, along with a decreased contribution from the [0, 12) and [18, 30) groups, may have sustained the dominant role of the [12, 18) and [30, 40) groups in transmission in late June 2021. Additional details are provided in the Supplementary Discussion and Supplementary Fig. 7. On June 23, 2021, the [30, 40) age group exhibited a remarkable elasticity of $\tilde{e}_{[30, 40), 25} = 0.36$, coinciding with an $R_t$ above 1. In July and August 2021, a consistent $S_{j,t} = 3$ across waves 26 to 29 highlights adults in the [18, 40) age group as the main contributors to transmission. Meanwhile, the [12, 18) age group reverted to a marginal role (see Table 2 and Fig. 5) and experienced a noticeable decline in susceptibility starting in late June 2021 (wave 25). However, the shift of age-specific elasticity during this period was primarily driven by changes in contact patterns, as discussed in detail in the Supplementary Discussion (Supplementary Fig. 6).

### 03/09/21-29/10/21, survey waves 30 to 34

September and October 2021 marked a phase of further relaxation of NPIs (see Fig. 6), with the stringency index dropping to 32 by October 2021. After school reopened in September, CoMix survey data from waves 30 to 34 revealed notable variations in daily contact numbers, particularly among individuals aged under 18 years. Children under 18 years old reported an average of 8.53–12.10 daily contacts, compared to an average of 4.86–5.93 for adults (see Table 2 and Supplementary Fig. 13). During this period, adult susceptibility, which had been rising since July, peaked between September and November 2021 (Fig. 4), driven by waning immunity and the increasing circulation of the Delta VOC[40]. These shifts are reflected in the cumulative sensitivity and elasticity indices, which highlight children under 12 and adults aged [18, 50) as the main contributors to SARS-CoV-2 transmission (Figs. 3 and 5). Interestingly, despite a high overall contact rate within the [12, 18) age group, its contribution to transmission remained marginal. This may be attributed to its highly assortative mixing pattern and lower susceptibility, which together likely limited its role (Fig. 3 and Supplementary Fig. 9b). At the same time, the return to school coincided with an unprecedented elasticity value of

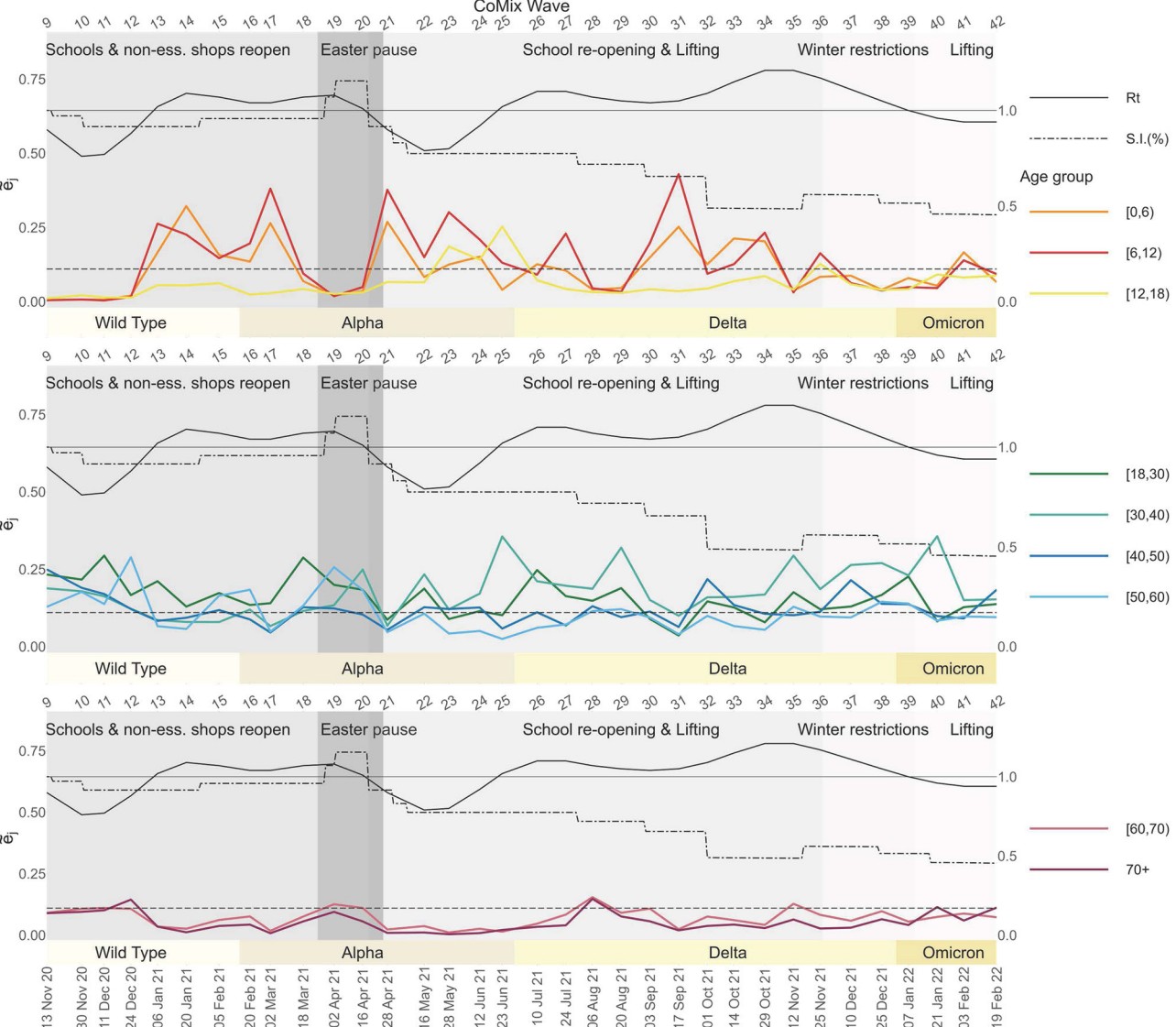

**Fig. 5 | Age-specific elasticity evolution: children (top), adults (middle), elderly (bottom).** On the top *x*-axis, we report the sequence of CoMix waves; on the bottom *x*-axis, the corresponding calendar date. On the *y*-axis, we display the corresponding value of the elasticity indices relative to each of the age groups considered, together with the stringency index[50] indicating the severity of the social distancing policies in place: here the index is rescaled such that the maximal level of stringency is 1. The dashed horizontal line marks the value 0.11 for the elasticity, i.e., average contribution to $R_t$. In addition, the effective reproduction number ($R_t$) is plotted in solid black and on a different scale indicated by the secondary *y*-axis (right); the solid horizontal black line indicates the critical threshold 1. A bar just above the main *x*-axis indicates the emerging VOCs of SARS-CoV-2. The length of every coloured bar represents the period of time during which the corresponding VOC was detected in more than 50% of the sequenced SARS-CoV-2 infections.

$\tilde{e}_{[0,6),31} = 0.43$, making children in the [0, 12) age group the main contributors to transmission on September 17, 2021 (wave 31), when $S_{j,t}$ peaked at 3 for $j = [0, 6)$ and [6, 12). From October 1, 2021 (wave 32), further easing of NPIs, including the reopening of pubs and dance clubs[51], corresponded with a shift in transmission dynamics. Adults aged [18, 50), particularly the [30, 40) group, had the highest proportional impact on $R_t$, joined later by children in the [0, 12) age group on October 29, 2021 (wave 34) (Fig. 3). Defining age-specific contributions to $R_t$ using cumulative elasticity is valuable, especially when $R_t$ exceeds 1, as it signals potential exponential growth. This was the case during this period, with $R_t$ peaking at 1.2 in late October and early November 2021. The analysis indicates that interventions targeting children under 12 years had a high potential to reduce $R_t$. For example, a hypothetical 14% reduction in the number of susceptibles in the [6, 12) age group at the start of the school year (wave 31) would have lowered $R_t$ just below the critical threshold of 1 (Supplementary Fig. 8).

### 12/11/21- 04/03/22, survey waves 35 to 42

The stringency of NPIs slightly increased between mid-November 2021 and early January 2022 due to new restrictions aimed at reducing the winter burden on the healthcare system, triggered by increased hospital admissions[51,52]. This led to a general reduction in social contacts, although individuals under 18 years of age maintained a relatively high level of contact until November 10, 2021 (wave 37) (Table 2). Concurrently, the Belgian government expedited administering a third vaccine dose, starting in early October 2021, to increase coverage among vulnerable and over-65-year-old individuals before Christmas. This is reflected in a gradual decline in age-specific curves of the susceptible population, counteracting the previously observed rebound (Fig. 4). Our perturbation analysis describes a notable change in SARS-CoV-2 transmission dynamics during this period. Children's contribution to transmission became marginal, while adults, especially those aged [30, 40), assumed a sustained dominant role. Table 2 shows a

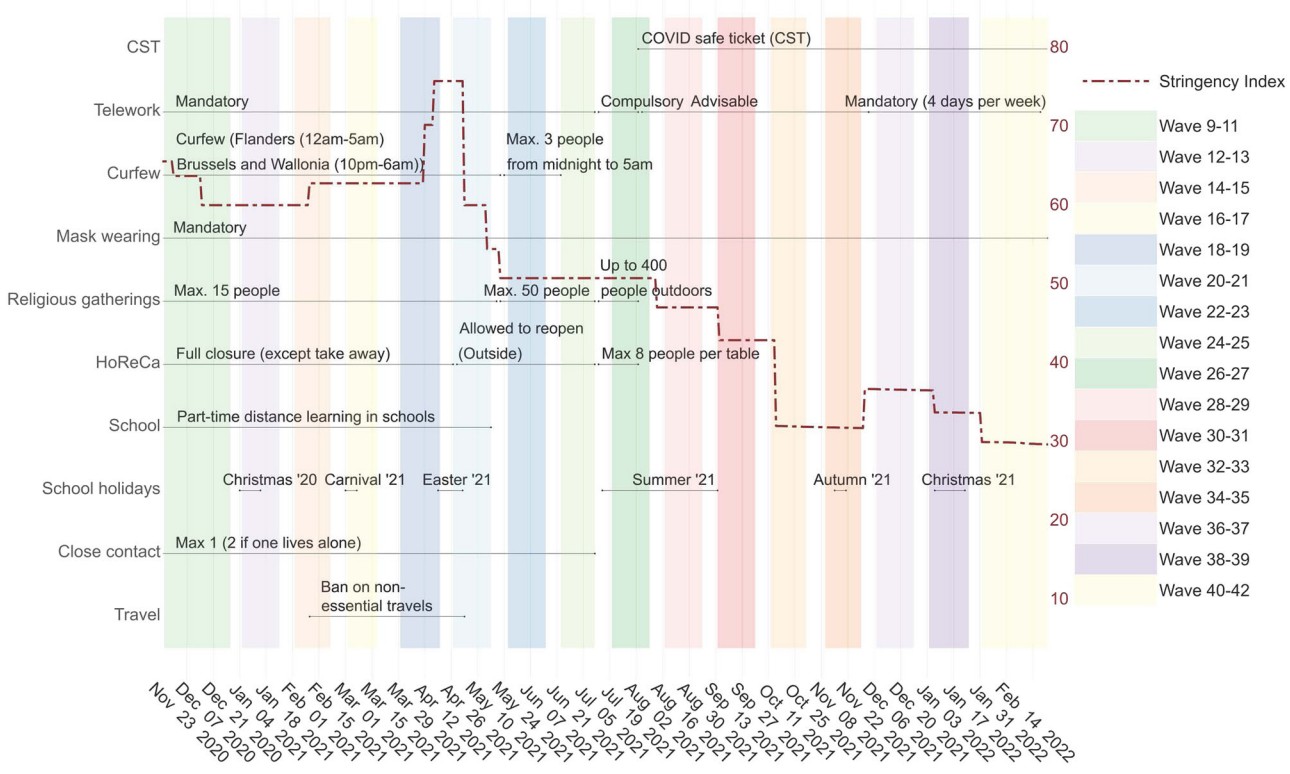

**Fig. 6 | Timeline of the main NPIs in place in Belgium throughout the observation period.** The red dashed line corresponds to the average stringency index[50].

consistent $S_{[30, 40),t} = 3$ during this period, highlighting the above-average cumulative elasticity of this group (Fig. 5) and sensitivity (Supplementary Fig. 14). At wave 40 (January 21, 2022), the Omicron VOC, characterised by increased transmissibility[44,45], was the dominant strain, accounting for 99.7% of sequenced cases[50,53]. At this time, the perturbation analysis identified adults aged [30, 40) as the main contributors to transmission. This is emphasised by a peak elasticity value of $\tilde{e}_{[30,40),40} = 0.36$. Figure 4 illustrates a marked decrease in the number of susceptibles from January 7, 2022 (wave 39), coinciding with a surge in infections across all age groups[49,52]. A notable exception is observed in adults over 60 years old, who received booster vaccinations early on. For this group, a linear increase in susceptibility coincides with the emergence of the Omicron VOC, possibly reflecting immunity waning combined with the immune evasion features of the variant[45,55].

## Discussion

This study examines how age-specific susceptibility, vaccination coverage, social behaviour, and NPIs shaped SARS-CoV-2 transmission dynamics in Belgium. By conducting a longitudinal perturbation analysis of the NGM, we provide insights into the role of different age groups over time, offering a valuable tool for targeted pandemic management.

In our study, children aged [0, 12) emerged among the main contributors to SARS-CoV-2 transmission at specific periods, particularly in early 2021 and September-October 2021, contrasting with their marginal role during the early pandemic stages[2,25,28]. The said periods coincided with school reopenings, high contact rates, and delayed vaccination efforts for young children, while other age groups benefited from prior immunity through infection or vaccination.

We emphasise that our analysis offers a relative measure of age-specific contributions to transmission, capturing the dynamic interplay between susceptibility, contact rates, and immunity at each observation point. Vaccination efforts targeting older populations can increase the relative susceptibility of children, amplifying their role in transmission when combined with high contact intensity. Even with stable overall susceptibility of younger groups, local factors such as immunity waning in the population or the emergence of VOCs can significantly enhance children's influence during specific periods.

Moreover, by the mathematical definition of the indices employed in our analysis, children emerged as potential optimal targets for intervention during specific moments. Their above-average cumulative sensitivity and elasticity indices suggested that changes in their epidemiological parameters were expected to produce larger fluctuations in $R_t$, particularly when it exceeded the threshold value of 1. This pattern was evident at the beginning of 2021, coinciding with the resumption of face-to-face education in primary and secondary schools. The observed fluctuation in the elasticity of children aged [0, 12) years coincided with shifts in contact patterns (Supplementary Figs. 3 and 13). A similar trend was noted between April 28 and June 12, 2021, following the relaxation of NPIs after Easter. However, during this period, the estimated $R_t$ based on confirmed cases remained safely below the value 1. Children again emerged as main contributors to transmission in September and October 2021, coinciding with another rise in $R_t$ above 1 (Fig. 3). Here, while their high daily contact rates were a factor, the evolving immunity landscape—shaped by the vaccination campaign and the circulation of new viral variants—also played a crucial role in redefining the transmission hierarchy among age groups (see Supplementary Fig. 9). Throughout the study period, high daily contact numbers reported by children aged up to 12 years often aligned with their increased contributions to virus transmission (as indicated by high elasticity values). However, this relationship is not always straightforward. For instance, the [12, 18) group reported consistently high contact rates but contributed marginally to transmission, except during late May-June 2021. During this period, increased disassortative mixing of this group with adults (Supplementary Fig. 7d) corresponded with infective values $\upsilon_{[12, 18),t}$ above average (Supplementary Fig. 15). This observation aligns with our mathematical characterisation of the dominant left eigenvector (**v**) components as the

gradient of $R_t$ with respect to the NGM's rows, which directly involve the group's contact structure. High infective values indicate age groups with a greater per-susceptible risk of initiating transmission chains, driven by this strong behavioural component. In addition, variations in age-specific susceptibility strongly influence changes in the NGM rows. We consistently observed higher-than-average $v_{j,t}$ values for the [0, 6) and [6, 12) age groups, underscoring the critical need for focused screening and contact tracing in younger age groups to effectively mitigate transmission, as supported by other research[11,29,46].

Understanding immunity changes within these age groups is crucial for controlling transmission dynamics. Our simulations indicated that before the vaccination campaign in December 2020, children under 18 had approximately half the susceptibility of adults, consistent with studies linking age-related susceptibility increases to stronger initial antibody responses in younger individuals[1,5,10,56,57]. Our sensitivity indices identified children in the [0, 12) age group among the main contributors to virus transmission between January and March 2021. This period coincided with the Alpha VOC becoming the dominant strain in Belgium, which has been associated with higher transmissibility and increased susceptibility in younger populations[36–38]. Similarly, this group of children exhibited an above-average proportional contribution to transmission between September and October 2021, when the Delta VOC—characterised by higher $R_t$, increased hospitalisations, and vaccine-evasive properties[40,41,58,59]—was the predominant circulating variant. By July 2021, vaccination in Belgium had been extended to adolescents aged 12 years and older, but younger children remained unvaccinated until early 2022, leaving them relatively more susceptible and influential in transmission dynamics. In the Supplementary Discussion (Supplementary Figs. 8 and 9), we apply perturbation analysis to demonstrate that earlier vaccination of the [0, 12) age group could have played a key role in reducing $R_t$, potentially helping to keep it below the value 1. However, during this period, while vaccines were confirmed to be safe for children aged 5 to 11 years[60,61], emerging data and public caution—partially fuelled by reports of adverse events[62,63]—contributed to delays in vaccination rollout for this age group.

Adults aged [18, 60) consistently exhibited high cumulative sensitivity indices, reflecting their larger pool of susceptibles and above-average q-susceptibility and q-infectiousness (Supplementary Table 3 and Supplementary Fig. 14). Within this group, young adults aged [18, 30) were among the main contributors to transmission for most of the observation period, particularly during the rise in hospitalisations observed in March 2021. This dynamic shifted with the introduction of stricter NPIs across Easter (March 31 to April 19), coinciding with a drop in $R_t$ below 1 and a decline in the [18, 30) group's cumulative elasticity, which fell below average for the first time by the end of April 2021 (Fig. 5).

The emergence of new VOCs, ongoing vaccination efforts, and the easing of NPIs—along with corresponding shifts in contact patterns—created an intricate landscape of transmission dynamics. Our analysis helped identify that, from late June 2021, as the Alpha strain was replaced by the Delta VOC, the [18, 40) age group played a crucial role in sustaining virus spread. Within this group, the [30, 40) age group emerged as a main contributor, a role that became even more pronounced by the end of December 2021 when the Omicron VOC replaced Delta as the dominant strain. Throughout this period, the [30, 40) group consistently exhibited high elasticity and infective values, surpassing younger age groups in their impact on transmission (see Fig. 5).

Our analytical approach leverages extensive data collected during the COVID-19 pandemic, capturing the evolving transmission dynamics from November 2020 to February 2022. The social contact data from this period provided the interaction network between age groups, while epidemiological parameter estimates determined the likelihood that each interaction resulted in a new infection. This framework shaped the role of each age group in overall transmission. Through our

perturbation analysis, we developed indices that capture the hierarchy of these roles, summarising how changes in contact patterns, NPI stringency, age-specific susceptibility, and $R_t$ influenced the contributions of different age groups to virus spread. At the same time, the somehow inverse analysis of pinpointing specific causes behind shifts in these indices is inherently challenging due to the constantly evolving NGM's structure. In the Supplementary Discussion, we present additional theoretical tools (detailed in the Supplementary Methods) that provide valuable insights into our case study and may serve as a basis for future research into these dynamics. Specifically, applying differential calculus to the sensitivity indices allows for a deeper examination of the factors driving age-related transmission patterns.

While our study offers an unexplored perspective on epidemic observation, several other studies have employed social contact data and the next-generation matrix approach to explore the effects of NPIs on age-specific transmission dynamics across various countries. In the UK context, multiple studies[64–66] have shown that young adults aged 20–50 years were the primary contributors to virus transmission overall. However, the role of children, particularly those aged 5–17 years, became increasingly important when schools were open. Notably, the reopening of schools after prolonged restrictions was identified as a key driver of transmission increases in these studies. Similarly, studies in the Netherlands[67] and Italy[68,69] found that reduced school contacts among children aged 0 to 17 years were pivotal in controlling virus spread during closures and driving transmission after reopening. In Canada[70] and South Korea[71], school closures, alongside social distancing, were highly effective in reducing transmission. In China[72,73], prolonged school closures kept the role of the [0, 18) age group minimal, with their relative contribution to virus transmission being much lower than that of the age group of adults [30, 60) during the spread of Omicron variants from March to June 2022. These studies support our findings and emphasise the critical need to accurately assess the impact of different age groups on transmission dynamics when designing control policies.

Finally, it is essential to acknowledge the limitations of our study, which may influence the interpretation and generalisability of our findings. One key limitation is the resolution of available data. Our age structure, aligned with Belgium's educational settings, relies on multiple data sources to inform age-specific epidemiological parameters[1,8,10]. However, in early epidemic stages, such detailed data may be unavailable[17], limiting the generalisability of our approach to other contexts and the possibility of reconstructing finer age structures from social contact surveys[74]. On the other hand, efforts to reconstruct social contact data on a finer time scale[75] would enhance the potential of our linear analysis[25,76]. This is particularly crucial when social contact patterns change rapidly, such as with new NPIs or seasonal shifts, an aspect not explicitly addressed in our study. Another limitation concerns data uncertainty and its impact on our results in a heterogeneous framework like ours. Reporting biases, sampling errors, and selection biases can affect the age-specific representativeness of social contact data[12]. A recent study emphasised the importance of accounting for the frequency and repetitiveness of contacts, as these factors can significantly influence epidemic dynamics[75]. Uncertainty in contact patterns, susceptibility estimates, and the parameters informing our NGM likely affect the statistical significance of detected age-specific differences. Future work should include comprehensive error propagation analysis while considering contact correlations within and between age groups[12,77]. Nevertheless, while our approach operates in a deterministic fashion by way of average values, it lays the foundation for future explorations employing perturbation analysis coupled with stochastic models. This creates a pathway for more robust, uncertainty-accounting studies, aligning our approach with evolving needs in pandemic response. The present study focuses on SARS-CoV-2 transmission in Belgium, though the flexibility of our methodology makes it applicable to a wide range of infectious diseases where transmission is tied to contact rates. This adaptable approach offers a valuable tool for

analysing disease dynamics and has broad relevance in epidemiology and public health management.

## Methods

### Overview

Our research expands upon the concept of NGM perturbation analysis[25] by conducting a longitudinal study. We examine dynamic shifts in the epidemiological landscape across distinct time points, from November 2020 to March 2022. The proposed approach allows us to monitor the changes in sensitivity indices over time. The time points in our study align with 34 consecutive waves of social contact data collection[24], gathered from a representative sample of the Belgian population. For in-depth information, refer to the "Social Contacts" section below. The analysis in our study is executed at each time point through a two-step procedure. Initially, we derive the next generation matrix[26] from an SEIR compartmental model[8,48], which was designed to capture the spread of SARS-CoV-2 in Belgium. Within this framework[26,27], the NGM is represented as a square matrix whose dimensions depend on the infectious states considered in the system, in this case, the age groups. Importantly, the spectral radius of the NGM indicates the outbreak potential, equivalent to the average number of secondary infections caused by a typical infected individual while infectious. In a fully susceptible population, this represents the basic reproduction number; otherwise, it reflects the effective reproduction number ($R_t$). The NGM in our study incorporates the latest data on age-specific social contacts, the count of susceptible individuals per age group, and the $R_t$ derived from available PCR test-based estimates[78]. Specifically, the effective reproduction number at each time step is the arithmetic mean of the daily $R_t$ estimates over a 7-day period centred on the date of the corresponding social contact survey wave. This ensures that the NGM's dominant eigenvalue matches the observed $R_t$ values, avoiding reliance on $R_t$ estimates directly derived from the SEIR model simulations. Importantly, this calibration choice does not affect our perturbation analysis, as the sensitivity indices defined in our study are invariant under the multiplication of the NGM (and the $R_t$) by a nonzero scalar.

The SEIR model further provides a robust framework for reconstructing the evolving age-specific susceptibility[8,48]. By accounting for various forms of protection, including natural infection, vaccination, and the impact of viral variants of concern (VOCs), it successfully integrates heterogeneous data sources—such as hospitalisation, seroprevalence, and PCR positivity data—to reliably capture epidemic trends in Belgium. In the second step, the epidemiological characterisation from the first step is employed to conduct a longitudinal perturbation analysis. This process enables us to analytically quantify how local variations in the parameters of the NGM entries influence the $R_t$. Changes in epidemiological parameters may be due to variations in NPIs, viral mutations, shifts in contact behaviour, depletion of susceptibles from natural infection progression, and the effects of vaccination campaigns.

### Social contact data

The core assumption underpinning the calculation of changes in infected host numbers due to virus transmission relies on the social contact hypothesis[79]. According to this hypothesis, the number of secondary infections an infected individual generates is proportional to their social contacts. The proportionality constant (indicated as $q$) and the definition of relevant contact hinge on the specific pathogen under consideration. A contact qualifies as either a face-to-face conversation of at least a few words or skin contact, aligning with the definitions used in the principal studies on social contacts in Belgium[7,24]. Our study employs social contact data derived from 34 consecutive waves of the CoMix survey conducted in Belgium from November 2020 to March 2022 amid the COVID-19 pandemic. The survey waves were collected every two weeks. Participants logged their daily contacts detailing the type, location, and age of the person

contacted. The data were subsequently processed and stratified by age using the open-source tool SOCRATES[23,24]. Social contacts shape the structure of the next-generation matrix, influencing its overall structure. From the CoMix survey, we obtain the average daily number of contacts that an individual of a particular age ($i$) makes with individuals of age ($j$), denoted by $m_{ij}$. This information helps us construct the pivotal social contact matrix. Subsequently, we process these matrices further to:

i.  Meet the reciprocity constraint. Given the nature of the contacts considered, the total number of contacts between two age groups ($i$ and $j$) should be the same whether derived from $m_{ij}$ or $m_{ji}$. In simple terms, $N_i m_{ij} = N_j m_{ji}$, where $N_i$ and $N_j$ are the number of individuals in each age group considered.

ii. Account for the impact of participation fatigue, particularly in longitudinal studies engaging participants over extended periods. A recent study[12] conducted on CoMix survey data collected in Belgium revealed that individuals participating in multiple survey waves tended to under-report the number of their daily contacts, indicating a potential impact of participation fatigue on the accuracy of the data collected. In the current research, we incorporate the adjustments suggested by the aforementioned study to correct for this bias. This involves adapting the social contact matrices according to the wave and age of the participants.

iii. Account for the impact of symptom onset on the infected individuals' level of social interaction, particularly in non-household environments. This acknowledges that symptoms typically reduce social contacts[80], notably influencing the virus transmission dynamics. We refer to the Supplementary Methods for further details.

### Model choices and calibration

Crucial to our analysis is the derivation of the NGM from the chosen compartmental model describing disease dynamics. We consider an age-structured Susceptible, Exposed, Infectious, and Recovered (SEIR) model developed by Abrams et al.[8], categorising individuals into ten age groups, each spanning ten years. To align with our specific age partitions, $\Omega = [0, 6), [6, 12), [12, 18), [18, 30), [30, 40), [40, 50), [50, 60), [60, 70), [70, \infty)$, we adjust these groups, assuming that the distribution within overlapping intervals reflects the demographic composition. This adjustment is particularly important for those under 18 years, mirroring divisions within the Belgian school system. This age structure is matched by age-specific estimates of key epidemiological parameters, such as q-susceptibility (the probability of a susceptible individual becoming infected after close contact) and q-infectiousness (the probability of an infected individual transmitting the virus during close contact). These quantities, as in Franco et al.[10], are defined up to a constant $q$ which is calibrated at each observation point so that the NGM dominant eigenvalue matches the value of $R_t$ estimated from positive PCR tests[78]. Transition rates through the different infectious states after exposure are also modelled according to this age structure (more details in the Supplementary Methods). The model tracks the progression from a susceptible state ($S$) to an exposed state ($E$) upon effective contact with an infectious person. After exposure, individuals enter a pre-symptomatic infectious state, followed by either a symptomatic or asymptomatic state before recovery. Symptomatic cases may progress to severe illness, potentially leading to hospitalisation or ICU admission. Although the model includes disease-related mortality in hospitalised cases, these factors do not affect the NGM structure. It is the choice of infected and infectious compartments and the specific age structure that defines the NGM[26]. The NGM's formulation depends on parameters governing transmission (the force of infection) and transitions between various infectious states (occupancy time in each state). Pre-infectious and post-infectious states are irrelevant to the NGM's formulation; hence, using a different model (e.g., SIR) with the same age structure and infectious state choices would yield an

identical NGM. However, it is important to consider the complete model structure when extending our perturbation analysis over time. Since we don't have comprehensive serological data for Belgium, we use numerical simulations to track the age-specific changes in the susceptible population. This, in turn, affects the structure of the NGM and, consequently, the sensitivity indices. To this end, we use the estimated susceptibility from an SEIRS model developed by Willem et al.[48], extending the original SEIR model[8] to include VOCs circulation, vaccination uptake, and waning immunity. The model uses a stochastic, binomial chain approach to simulate epidemic progression in discrete generations, with a time resolution of 0.25 days. It specifically accounts for VOC-related changes in transmissibility, hospitalisation risk, and latency period. The model also uses the same social contact data from CoMix surveys to estimate age-specific proportionality factors that translate reported contact rates into transmission rates (under the social contact hypothesis[79]). Susceptibility in our framework is based on four distinct immune states representing combinations of infection history and vaccination status, as outlined by Willem et al.[48] and better detailed in the Supplementary Methods. Individuals are considered susceptible if they have never been infected or vaccinated or have waning immunity. The model transitions individuals from full protection to susceptibility over time, reflecting the gradual decrease in immunity (see Supplementary Table 4). This approach ensures that susceptibility estimates reflect the dynamic nature of population immunity during the pandemic. To effectively integrate various data sources that became available at different stages of the pandemic in Belgium, the model calibration employed a multi-step Bayesian approach with Markov Chain Monte Carlo (MCMC) sampling using 60 chains. Each step focused on different time horizons and specific parameters, which were sequentially updated based on previous calibration results. Transmission-related parameters were initially estimated using data from hospital admissions[49], early seroprevalence data (available up to October 17, 2020)[81], and genomic surveillance data for Belgium[53]. Subsequently, parameters affecting hospital and ICU occupancy were estimated by minimising a least squares criterion, ensuring the best fit between observed and simulated hospital loads. Lastly, mortality-related parameters were refined to account for COVID-19-related deaths in hospitalised patients. From 60 MCMC chains, the 40 best-fitting parameter sets were selected based on their agreement with observed data. For each of these sets, 10 stochastic realisations were performed, resulting in 400 estimates of age-specific susceptibility over 730 days (March 1, 2020, to February 28, 2022). The daily age-specific number of susceptible individuals was then calculated as the mean across these estimates. A comprehensive explanation of the model, the calibration process and the result of the calibration are provided in Willem et al.[48], especially within the related Supplementary Information.

### Next generation matrix analysis

In our analysis, we utilise sensitivity indices to elucidate the roles of different age groups in SARS-CoV-2 transmission dynamics. These indices measure the impact of epidemiological changes on the effective reproduction number ($R_t$), as resulting from the next generation matrix (NGM). They are grounded in the concept of classical sensitivity index ($\partial R_t / \partial k_{ij} = s_{ij}$), which assesses the rate of change in the NGM's spectral radius due to a variation in a single matrix entry[76]. The epidemiological implications of these indices are further explained, and their mathematical foundations are detailed in the Supplementary Methods.

Key indices include:

- Cumulative Sensitivity ($\tilde{s}_{j,t}$): This index measures the impact on $R_t$ resulting from changes in how a single index case in age group $j$ transmits the infection, at time $t$. Higher $\tilde{s}_{j,t}$ values signify a greater sensitivity of $R_t$ to secondary infections from a single infected individual in age group $j$, thereby identifying effective spreaders in the current infected generation. Each index $\tilde{s}_{j,t}$ is

proportional to the relative incidence of the infection in age group $j$, as elaborated in the Supplementary Methods.
- Cumulative Elasticity ($\tilde{e}_{j,t}$): This index quantifies the proportional contribution to $R_t$ of an age group $j$ in the current generation (assumed to start at time $t$). High indices pinpoint age groups that substantially contribute to the overall disease propagation and for which proportional variations in the epidemiological parameter set translate into higher proportional shifts of $R_t$.
- Infective Value ($v_{j,t}$): This index quantifies the impact on the $R_t$ following a perturbation in either the force of infection exerted on individuals in age group $j$ or their susceptibility, at time $t$. Each $v_{j,t}$ corresponds to the influence a single new case in age group $j$ has on the number of new infections in each future generation. Specifically, one new case in age group $j$ is expected to increase the overall infection count by $R_t v_{j,t}$ in the next generation of infections, and by $R_t^m v_{j,t}$ after $m$ generations[25]. Higher values identify age groups with a higher potential for initiating transmission chains.

A composite index, $S_{j,t}$, combines the above indices to provide a comprehensive view of each age group's contribution ($j$) to virus propagation at specific times ($t$), aligned with CoMix survey waves. This index is derived as the sum of 3 Boolean values, specifically:

1. $\tilde{e}_{j,t} > \frac{1}{n}$;
2. $\tilde{s}_{j,t} > \tilde{s}_{avg}$ Or $v_{j,t} > v_{avg}$;
3. $\tilde{s}_{j,t} > \tilde{s}_{avg}$ & $v_{j,t} > v_{avg}$.

Condition 1 indicates that the age group $j$'s contribution to $R_t$ (as represented by $\tilde{e}_{j,t}$) exceeds 11%, corresponding to the average cumulative elasticity value with $n = 9$ age groups ($\sum_j \tilde{e}_{j,t} = 1$, see Supplementary Methods). Conditions 2 and 3 assess whether the cumulative sensitivity index $\tilde{s}_{j,t}$ and the infective value index $v_{j,t}$ surpass their respective average values, $\tilde{s}_{avg}$ and $v_{avg}$, calculated as the mean of the arithmetic averages of these indices across all survey waves. Throughout the 34 CoMix waves, we note $\tilde{s}_{avg} = 1(0.98, 1.03)$ and $v_{avg} = 0.93(0.90, 0.95)$, indicating the 99% confidence intervals. The resulting $S_{j,t}$ index, varying from 0 to 3, allows us to deduce the relative transmission roles of different age groups at specific times. A value of $S_{j,t} = 3$ indicates a distinctly above-average transmission role, as seen in Table 2. More specifically, individuals in such a high-index age group $j$ are primary contributors to overall virus transmission, likely to drive notable variations in $R_t$ upon experiencing age-specific epidemiological changes (such as through NPIs or vaccination), and possess a higher potential to initiate transmission chains when exposed to infection risks.

### Reporting summary

Further information on research design is available in the Nature Portfolio Reporting Summary linked to this article.

## Data availability

The datasets analysed in the current study are available in the Zenodo-based repository[82], https://zenodo.org/records/10549953, as well as through the CoMix-Socrates App https://socialcontactdata.org/tools/.

## Code availability

The analysis and results presented in this study are fully reproducible using the R code provided in the following GitHub repository: https://doi.org/10.5281/zenodo.14777392[83]. All scripts were developed in R version 4.3.0 (2023-04-21 ucrt).

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

## Acknowledgements

L.A., P.B., and N.H. acknowledge funding from the European Union's Horizon 2020 research and innovation programme – project EpiPose (Grant agreement number 101003688) and the ESCAPE project

(101095619), both funded by the European Union. N.H. further acknowledges support from the VERDI project (101045989), also funded by the European Union. Views and opinions expressed are, however, those of the author(s) only and do not necessarily reflect those of the European Union or the European Health and Digital Executive Agency (HADEA). Neither the European Union nor the granting authority can be held responsible. L.A. and N.H. acknowledge funding from the Special Research Fund through the Methusalem project BOF08M01 - phase III. C.P.C. acknowledges the funding by Fundação para a Ciência e a Tecnologia (FCT.BD). L.W. and S.A. acknowledge support from the Research Foundation Flanders (FWO) (ACCELERATE project G059423N). CoMix data collection in Belgium was made possible through funding from the European Union's Horizon 2020 research and innovation programme - project EpiPose (Grant agreement number 101003688), and with financial support from the National Public Health Institute of Belgium, Sciensano and Janssen Pharmaceuticals. This work reflects only the authors' view. We extend our gratitude to the research teams at the University of Antwerp, Hasselt University, and the London School of Hygiene and Tropical Medicine, involved in the CoMix study within the EpiPose project, for their invaluable contribution in designing the survey and processing, curating, and collecting the social contact data utilised in our study.

## Author contributions

L.A. contributed to conceptualisation, methodology, software development, formal analysis, original draft preparation, visualisation, manuscript review and editing, and project administration. C.P.C. was involved in conceptualisation, methodology, and manuscript review and editing. P.C., G.M., and S.A. provided resources and participated in manuscript review and editing. N.F., C.F., and P.B. reviewed and edited the manuscript. L.W. contributed to software development, conceptualisation, resources, and manuscript review and editing. N.H. contributed to conceptualisation, methodology, resources, original draft preparation, supervision, funding acquisition, project administration, and manuscript review and editing.

## Competing interests

The authors declare no competing interests.
