## [Transparent Peer Review file · Nature Communications]

Assessing the Role of Children in the COVID-19 Pandemic in Belgium Using Perturbation Analysis

Corresponding Author: Mr Leonardo Angeli

Version 0:

Reviewer comments:

Reviewer #1

(Remarks to the Author)

In this manuscript, Angeli et al. leverage longitudinal social contact surveys performed across 34 waves between November 2020 and February 2022 in Belgium to characterize shifts in the role played by different age groups in SARS-CoV-2 transmission during the pandemic. I find the perturbation analysis and the indices introduced by the authors to be particularly elegant and an effective way to summarize and analyze next-generation matrices. I think this approach could be valuable to understand the determinants of disease transmission in epidemiological settings where social contact data and information regarding groups' susceptibility can be reconstructed. I have some comments that I hope the authors will find helpful:

Major comments:

1. Model calibration

In this work, the reconstruction of age-dependent susceptibility across time is a critical step and can highly impact conclusions (by influencing the derivation of the next-generation matrix). Too little detail is provided regarding this important step. How are the calibration and simulation steps performed? Which data sources are used for the calibration? What is the quality of the fit to the calibration data? The fact that the authors report the increased risk in hospitalization for the different variants suggests that hospitalization data are used in their simulation step to reconstruct susceptibility but how? On which time period is the calibration performed? How are changes in transmissibility over time accounted for / calibrated? Are the Comix surveys accounted for in the reconstruction of population susceptibility? It is currently unclear how the reconstruction of susceptibility over time is performed.

Accurately capturing and reconstructing (i) the evolution of immune protection (both post infection and vaccination), (ii) disease incidence in different age groups and (iii) age varying q -susceptibility and infectiousness is an important step to reconstruct susceptibility profiles over time and to estimate the NGM's coefficients. It appears important to evaluate how well calibrated the model is. Have the authors perform some calibration check? For example, do the model simulations reproduce the incidence of COVID cases in different age groups throughout the pandemic? Somehow related, it would also be helpful to have a time series of cases by age across time in the different age groups.

2. Interpretation of the different sensitivity indices

The sensitivity and elasticity values reported for the different age groups are impacted by both behavioral (e.g. contact patterns) and biological factors (e.g. susceptibility) and current analyses do not enable to disentangle the mechanistic drivers of changes in groups contribution to transmission. Some of the interpretations provided in the results nonetheless suggest a causative relationship between some changes in susceptibility / contact intensity and the contribution of age groups to transmission even though the analytical framework does not currently support this. This is for example the case in (but not only):

- Line 152-153: "This shift corresponds to their high number of per capita contacts, which translated into a high infected values"

- Line 229: "The role of children age [0,12) notably fluctuated in correlation with their average number of daily contacts"

Some of the writing currently suggest some causation between changes in susceptibility / contact patterns and sensitivity/elasticity indices even though the current approach does not enable to disentangle these different drivers. I would suggest at the very least rephrasing the results to make it very clear that this is currently not a causation analysis but simply observation of a temporal correlation between e.g. changes in susceptibility and an increase in a sensitivity index.

3. Relative role played by immune vs behavioral factors

Following up on point 2, understanding whether changes in groups' contribution to transmission are driven by vaccination, infection or changes in contact patterns are very important and the results currently read as a (long) description of changes in sensitivity indices in Belgium over time without providing insights into the underlying mechanistic and dynamic drivers. I imagine that this could easily be explored using the authors' perturbation analysis framework by exploring different counter-factual scenarios. For example, to understand how changes in immunity vs contact patterns are impacting group contribution to transmission between wave W-1 (or W-2, W-3...) and W, one could compute counter-factual NGMs as follows:

- A NGM only accounting for changes in contact patterns between W-1 and W (using susceptibility profiles from wave W-1 but the Comix survey from wave W to compute the NGM)
- A NGM only accounting for changes in contact patterns between W-1 and W (using susceptibility profiles from wave W but the Comix survey from wave W-1)

Assuming this is doable, I really think this would strengthen the points made by the authors around the value of their perturbation framework to understand the role played by different age groups. I don't think it would be necessary to do it for every waves, but including a couple of these extra analyses (if relevant) might also help address some of the comments I had in 2.

Minor comments:

1. The term "sensitivity analysis" is widely used in the field to denote analyses evaluating the impact of changing a parameter value on the outcome (as a way to quantify the role played by different parameters on the expected outcome). Reading the title, this is initially what I thought the authors would be doing in this manuscript. Replacing "sensitivity analysis" by "perturbation analysis" (or generally another term that the authors might find more relevant) could help clarify this.
2. Line 33-35: The sentence is hard to understand. Splitting it in two would help clarify: "It integrates several age-specific factors [...] and demographics. It models transmission by [...]."
3. The concept of NGM is key in this study. It would be beneficial to give more details about what the NGM represents early in the introduction (when it is introduced). At the moment, the introduction of this concept is pretty quick, where a perturbation analysis of the NGM is mentioned early one without giving much context about why this matrix is important and what it measures.
4. Line 70-71: Please provide a brief description of the insights gained by these different indices. The methods are well described in the supplements and in the methods section, but it would be helpful for the readers to be able to understand what these key indices represent in the main text. This could for example (but not necessarily) include a small schematic of what a high / low sensitivity and a high / low elasticity represent.
5. Figures look great but it would be helpful to increase the font size for readability.
6. Many of the notations used in SM3 to describe the model structure are not defined (model compartments' names and meaning and some of the parameters used in the ODEs and the derivation of the NGM's coefficients).
7. Line 178: Could you please clarify what is meant by: "underscoring the relevance of the sensitivity analysis results"?
8. I appreciate that the authors accounted for age-varying q -susceptibility and q -infectiousness as this seems like an important element to account for (especially when considering the role played by children in SARS-CoV-2 transmission). I would recommend mentioning it very early on in the main text (it took me some time to find that information in the SM and I think this is important enough to be mentioned in the main text, when introducing the framework).
9. I understand that incorporating uncertainty might be beyond the scope of this paper, but it would be helpful to understand how population susceptibility estimations vary across replicate (this might just be adding a shaded area with uncertainty in Figure 6). This would be helpful to understand how reasonable the step of considering the mean across the 400 simulations is.
10. Line 118: "In response to a new peak in weekly hospitalization and a sustained rise in new infections, notably among individuals aged 10 to 19 years and mainly originating from school and work settings, Belgium implemented more stringent NPIs at the end of March." I was confused on how to interpret this comment with respect to the fact that S_j^t indices are low in individuals aged [0,18) during wave 18 and 19. Wouldn't we expect this indices to be high in people aged 10-19 if infections are high in that groups (especially with assortativity in mixing between age groups)?
11. The code links to a GitHub repository associated with another related manuscript and not to this manuscript.

(Remarks on code availability)

The code links to a GitHub repository associated with another related manuscript and not to this manuscript. The README does not contain any information on how to run the code.

At a first glance of the GitHub repository, beyond the fact that this is not the one associated with this manuscript, more documentation within the R scripts to understand what the code is doing would be helpful.

Reviewer #2

(Remarks to the Author)

While this manuscript presents original work and is worthy contribution to the literature, it is simply too long (>10000 words) in the current format.

The authors should make a choice of the aspects of their work they want to present, decrease the size of introduction and discussions. The results section should only present results and not reiterate the methods.

The manuscript has a very Belgian focus and doesn't discuss results in an international context. Have other research groups found similar results? Finally they did not discuss whether a similar analysis could be done using other data sources, e.g. from testing data or contact tracing data, which would confirm the findings that different age groups were more important during certain phases of the pandemic.

(Remarks on code availability)

Reviewer #3

(Remarks to the Author)

This paper proposes a novel approach that focuses on the changing roles of different age groups in the transmission dynamics of SARS CoV-2 in Belgium from November 2020 to February 2022. The proposed method highlights the distinct roles played by various age groups in the virus spread.

Strengths:

- + The paper is well-written and easy to follow, and the problem setting and discussions are all clearly described.
- + Four different roles in transmission and survey waves for various age groups are well-presented.
- + The effectiveness of the proposed approach is supported by empirical results.
- + The contributions are significant and interesting for the epidemics and epidemiology domain.

Weaknesses:

- The proposed approach is based on the SEIR model. Ideally, if authors could demonstrate results and conclusions by using SIR, SEIS, and etc, that would be even better.
- The proposed approach from the paper is not included in the comparison study in the evaluation.
- The results need further explanation. For instance, why some mean R_t values are larger than 1 but some are less than 1 (for different waves) in Table 1?
- It would be helpful if the authors can provide the reasons why using Bayesian approach with MCMC sampling to calibrate the model.

(Remarks on code availability)

I checked the R codes added to the Github repository. They looks good. It would be good if the authors can add more details about implementations in the README file.

Version 1:

Reviewer comments:

Reviewer #1

(Remarks to the Author)

Thank you for the revised manuscript, where many elements have been clarified.

There are many instances where values indicated in the main text don't match what is reported in the figures (e.g. line 96, line 114, line 158) This makes it really difficult to evaluate the consistency and the interpretability of the results. The manuscript needs a thorough read with a rigorous checking values included in the text and figures. I did not do a detailed investigations of inconsistencies, but please ensure that both the ones I just listed and potential others are corrected.

The study concludes to children as being "key contributors to SARS-CoV-2 transmission at various points throughout the observation period". First, the term "key contributor" is pretty broad and I think it would be helpful to specify what this term refers to (line 193). Second, I don't really understand how to interpret this finding with regards to the fact that susceptibility remains very high in children until the end of 2021. If they were key contributors to the epidemics, shouldn't they be infected? It feels difficult to conclude that they are such key contributors (if they aren't infected, they can't spread the virus either).

I still find certain paragraphs the methods section difficult to read though I appreciate the improvements there. The role of the SEIR like model vs R_t estimates from PCR data is unclear. Is the SEIR model only used to infer susceptibility over time? What is the R_t from PCR data used for? Only to scale the NGM and avoid relying on R estimates obtained from the SEIR model?

Line 267-269: Please clarify the paragraph comparing the results to other studies. The phrasing is currently confusing regarding the relationship between school closures and changes in transmission intensities.

I appreciate the new analyses with the counterfactual scenarios. Some legends are missing regarding the corresponding figures (S3 to S6). I imagine the dotted coloured line represent the non-counterfactual scenario but this is not indicated. There are many elements on this graph (two axes, dotted and plain black line, dotted and plain coloured lines...), and it's not obvious which axis is used to depict what and what the different elements indicate.

Finally, I did not understand how the elasticity gradient should be interpreted as. Please provide guidance on how to use that index and the interpretation. This is important, especially as the authors are underlining the potential for their methods to be used in other contexts!

(Remarks on code availability)

Reviewer #3

(Remarks to the Author)

All my concerns are addressed based on the authors' rebuttal file. I recommend for acceptance.

(Remarks on code availability)

Yes, the Github repository has provides details of implementations.

Response to Referees:

"Insights into the role of children in the COVID-19 pandemic in Belgium: a longitudinal perturbation analysis"

Authors: Leonardo Angeli, Constantino Pereira Caetano, Nicolas Franco, Pietro Coletti, Christel Faes, Geert Molenberghs, Philippe Beutels, Steven Abrams, Lander Willem, Niel Hens

Dear editor,

The authors sincerely thank the reviewers for their thoughtful and constructive feedback, which has greatly contributed to enhancing the comprehensiveness and clarity of our manuscript. The major comments and requests led to a substantial restructuring of the manuscript, significantly improving its readability and overall quality. We are confident that the revised version thoroughly addresses the critiques raised. Below, we provide a point-by-point response (with our reply in blue) detailing the specific changes made in response to each comment.

Kind regards,
The authors

Reviewer #1

Major comments:

Comment #1. Model calibration

In this work, the reconstruction of age-dependent susceptibility across time is a critical step and can highly impact conclusions (by influencing the derivation of the next-generation matrix). Too little detail is provided regarding this important step. How are the calibration and simulation steps performed? Which data sources are used for the calibration? What is the quality of the fit to the calibration data? The fact that the authors report the increased risk in hospitalization for the different variants suggests that hospitalization data are used in their simulation step to reconstruct susceptibility but how? On which time period is the calibration performed? How are changes in transmissibility over time accounted for / calibrated? Are the Comix surveys accounted for in the reconstruction of population susceptibility? It is currently unclear how the reconstruction of susceptibility over time is performed.

Accurately capturing and reconstructing (i) the evolution of immune protection (both post infection and vaccination), (ii) disease incidence in different age groups and (iii) age varying q -susceptibility and infectiousness is an important step to reconstruct susceptibility profiles over time and to estimate the NGM's coefficients. It appears important to evaluate how well calibrated the model is. Have the authors perform some calibration check? For example, do the model simulations reproduce the incidence of COVID cases in different age groups throughout the pandemic? Somehow related, it would also be helpful to have a time series of

cases by age across time in the different age groups.

Response:

The comment highlights a crucial aspect of our study and has allowed us to clarify the choices and methodology underpinning the numerical simulation of susceptibility. In response, we have consolidated the original Sections "Population Susceptibility" and "Compartmental Model" into a single section entitled "Model Choices and Calibration." This new section addresses the questions raised in the comment, which we agree are of utmost importance. For the sake of conciseness and given the focus of this paper, we initially chose to provide only a brief overview of the model structure, assumptions, and parameters. We have now included a clear reference to the recent study by Willem et al. (2024), which introduces the stochastic compartmental model in detail and extensively covers the calibration process. The said study was not published at the time of our initial submission, and we apologize for any inconvenience this may have caused. Moreover, visual inspection of the performance of the calibration process in terms of model fit with regard to the observed hospitalizations, hospital load and mortality can be found in the Supplementary Material corresponding to the previously mentioned manuscript, Figure S4. However, we ensured that key aspects related to the estimation of the susceptibility profile are clearly presented in the revised version of the manuscript, so that the reader not necessarily need to refer to Willem et al. Precisely, Lines 360-376 of the main text summarise the methodology and data sources used in calibrating the simulation model parameters.

Comment# 2. Interpretation of the different sensitivity indices

The sensitivity and elasticity values reported for the different age groups are impacted by both behavioral (e.g. contact patterns) and biological factors (e.g. susceptibility) and current analyses do not enable to disentangle the mechanistic drivers of changes in groups contribution to transmission. Some of the interpretations provided in the results nonetheless suggest a causative relationship between some changes in susceptibility / contact intensity and the contribution of age groups to transmission even though the analytical framework does not currently support this. This is for example the case in (but not only):

- Line 152-153: "This shift corresponds to their high number of per capita contacts, which translated into a high infected values"

- Line 229: "The role of children age [0,12) notably fluctuated in correlation with their average number of daily contacts"

Some of the writing currently suggest some causation between changes in susceptibility / contact patterns and sensitivity/elasticity indices even though the current approach does not enable to disentangle these different drivers. I would suggest at the very least rephrasing the results to make it very clear that this is currently not a causation analysis but simply observation of a temporal correlation between e.g. changes in susceptibility and an increase

in a sensitivity index.

Response:

We sincerely thank the reviewer for this insightful comment. We agree that it is crucial to avoid implying causality between the socio-epidemiological perturbations and the variations in transmission patterns without the analytical and methodological support to do so. As rightly pointed out, our analysis does not focus on the mechanistic drivers behind changes in the indices used. To address this, we have carefully revised the manuscript to ensure that we convey contemporaneity or association rather than causality. Furthermore, we have provided more context when discussing specific changes in transmission dynamics (please refer to our response to Comment #3 for further details).

In response to the specific instances mentioned:

- The original lines 152-153 have been replaced with a more nuanced explanation found between lines 130-137 in the revised manuscript:

“During May-June 2021 (waves 22 to 25), individuals aged [12,18) showed high elasticity and an increasing role in transmission, contrasting their marginal role until late April (wave 21). Their growing contribution cannot be solely explained by the increase observed in daily contacts in the same period (Supplementary Figure 13); elasticity patterns at waves 14 (Figure 1b) and 31 (Figure 3a) provide a good counterexample, showing their marginal role despite high contacts. Further analysis of the cumulative elasticity gradient with respect to the NGM’s columns suggests that a more disassortative mixing and the decrease in the contribution of the [0,12) and [18,30) groups, may have sustained the dominant role of the [12,18) and [30,40) groups in transmission during wave 25. Additional details can be found in the Supplementary Discussion and Supplementary Figure 7.”

- The original line 229 has been revised to the following (line 197-198):

"The observed fluctuation in the elasticity of children aged [0,12) years coincided with shifts in contact patterns (Supplementary Figures 3 and 13)."

Comment #3 Relative role played by immune vs behavioral factors

Following up on point 2, understanding whether changes in groups’ contribution to transmission are driven by vaccination, infection or changes in contact patterns are very important and the results currently read as a (long) description of changes in sensitivity indices in Belgium over time without providing insights into the underlying mechanistic and dynamic drivers. I imagine that this could easily be explored using the authors’ perturbation analysis framework by exploring different counter-factual scenarios. For example, to

understand how changes in immunity vs contact patterns are impacting group contribution to transmission between wave W-1 (or W-2, W-3...) and W, one could compute counter-factual NGMs as follows:

- A NGM only accounting for changes in contact patterns between W-1 and W (using susceptibility profiles from wave W-1 but the Comix survey from wave W to compute the NGM)
- A NGM only accounting for changes in contact patterns between W-1 and W (using susceptibility profiles from wave W but the Comix survey from wave W-1)

Assuming this is doable, I really think this would strengthen the points made by the authors around the value of their perturbation framework to understand the role played by different age groups. I don't think it would be necessary to do it for every waves, but including a couple of these extra analyses (if relevant) might also help address some of the comments I had in 2.

Response:

We greatly appreciate this comment, as it has allowed us to explore and demonstrate the potential of our approach further and to enhance the understanding of the complex dynamics that determine the contribution of different age groups to transmission. Following the reviewer's suggestion, in the Supplementary Discussion we discuss the results of several counter-factual exercises during key moments within the study period. These scenarios are further analysed using a second-order perturbation analysis, a mathematical tool that can shed light on the variations in the indices used in the study. These additional tools and analyses are presented and discussed in the Supplementary Methods (section "Second-Order Derivatives") and in the Supplementary Discussion. The insights derived from these analyses are referenced in the main text at Lines 117, 157, 233-234, and 257-259, and provide a deeper understanding of the factors driving significant changes in transmission patterns.

Minor comments.

1. The term "sensitivity analysis" is widely used in the field to denote analyses evaluating the impact of changing a parameter value on the outcome (as a way to quantify the role played by different parameters on the expected outcome). Reading the title, this is initially what I thought the authors would be doing in this manuscript. Replacing "sensitivity analysis" by "perturbation analysis" (or generally another term that the authors might find more relevant) could help clarify this.

Response:

We appreciate the reviewer's observation, which allowed us to reflect on the most suitable terminology to avoid potential misunderstandings. In our manuscript, we use the term *sensitivity analysis* to refer to the exploration of how changes in specific parameters or parameter sets affect outcomes—a broad definition that aligns with our work, as our

methodology examines how variations in NGM entries influence R_t . However, we acknowledge that *sensitivity analysis* is used to describe a variety of exercises, with a common example being the testing of robustness against changes in model assumptions. To enhance clarity, we have replaced *sensitivity analysis* with *perturbation analysis* throughout the manuscript when discussing the broader methodology. The rationale for this terminology is clarified in Lines 35-39 of the revised manuscript. The new manuscript title also reflect our choice.

2. Line 33-35: The sentence is hard to understand. Splitting it in two would help clarify: “It integrates several age-specific factors [...] and demographics. It models transmission by [...].”

&

3. The concept of NGM is key in this study. It would be beneficial to give more details about what the NGM represents early in the introduction (when it is introduced). At the moment, the introduction of this concept is pretty quick, where a perturbation analysis of the NGM is mentioned early one without giving much context about why this matrix is important and what it measures.

Response:

To address comments #2 and #3, we have revised part of the Introduction to improve clarity and effectiveness. Specifically, between lines 29 and 34 of the revised main text, we have provided a more comprehensive explanation of the NGM and its significance in our study.

4. Line 70-71: Please provide a brief description of the insights gained by these different indices. The methods are well described in the supplements and in the methods section, but it would be helpful for the readers to be able to understand what these key indices represent in the main text. This could for example (but not necessarily) include a small schematic of what a high / low sensitivity and a high / low elasticity represent.

Response:

To enhance the clarity and accessibility of our study's results, we have incorporated a brief interpretation of the key indices used in the analysis. This is now presented in Table 1, providing readers with a quick reference to understand what these indices represent, including what high values indicate.

5. Figures look great but it would be helpful to increase the font size for readability.

Response:

We thank the reviewer for the feedback. We increased the font size for higher readability.

6. Many of the notations used in SM3 to describe the model structure are not defined (model compartments' names and meaning and some of the parameters used in the ODEs and the derivation of the NGM's coefficients).

Response:

We appreciate the reviewer's comment, which emphasized the need for clearer definitions of the model compartments and parameters. In response, we have rewritten the "Compartmental Model" section in the Supplementary Methods (formerly SM3) to provide detailed definitions of the model compartments and a thorough explanation of the parameters used in the corresponding ODE system. This updated section ensures clarity and consistency in the notation used throughout the model structure.

7. Line 178: Could you please clarify what is meant by: "underscoring the relevance of the sensitivity analysis results"?

Response:

We rephrased the sentence between lines 162-164, namely:

"The use of cumulative elasticity to define age-specific contributions to R_t is crucial, especially when R_t exceeds 1, signalling potential exponential growth, as seen during this period with a peak of 1.2 in late October to early November 2021."

8. I appreciate that the authors accounted for age-varying q -susceptibility and q -infectiousness as this seems like an important element to account for (especially when considering the role played by children in SARS-CoV-2 transmission). I would recommend mentioning it very early on in the main text (it took me some time to find that information in the SM and I think this is important enough to be mentioned in the main text, when introducing the framework).

Response:

We thank the reviewer for this comment, which highlights an aspect of our model that was not adequately presented in the initial manuscript. We have now included these definitions earlier in the Methods section, specifically in the paragraph on Model Choices and Calibration (Lines 342-345) in the revised main text.

9. I understand that incorporating uncertainty might be beyond the scope of this paper, but it would be helpful to understand how population susceptibility estimations vary across replicate (this might just be adding a shaded area with uncertainty in Figure 6). This would be helpful to understand how reasonable the step of considering the mean across the 400 simulations is.

Response:

We appreciate this valuable suggestion. To enhance the interpretation of our results, we have added shaded areas representing uncertainty around the estimates in Figure 6, as recommended. While a full analysis of the impact of uncertainties is beyond the current scope, we recognize its importance in testing the robustness of our approach and have highlighted this as a limitation to be addressed in future work within the Discussion Section. (Lines 279-285).

10. Line 118: "In response to a new peak in weekly hospitalization and a sustained rise in new infections, notably among individuals aged 10 to 19 years and mainly originating from school and work settings, Belgium implemented more stringent NPIs at the end of March." I was confused on how to interpret this comment with respect to the fact that S_j^t indices are low in individuals aged [0,18) during wave 18 and 19. Wouldn't we expect this indices to be high in people aged 10-19 if infections are high in that groups (especially with assortativity in mixing between age groups)?

Response:

We appreciate the reviewer's sharp observation, which pointed out a discrepancy in the reporting of results. Upon review, we found that while there was a generalized increase in infections among school-aged individuals, the peak noted by the referenced study was particularly marked for children in grades 1 to 6 (aged 6-11 years) by the end of February. Indeed, the S_j^t index peaked at 3 for young children at wave 17. We have rephrased the paragraph (previously at Line 118) and clarified this point at the end of the preceding paragraph (Lines 95-98 in the revised manuscript).

11. The code links to a GitHub repository associated with another related manuscript and not to this manuscript.

Reviewer #1 (Remarks on code availability):

The code links to a GitHub repository associated with another related manuscript and not to this manuscript. The README does not contain any information on how to run the code. At a first glance of the GitHub repository, beyond the fact that this is not the one associated with this manuscript, more documentation within the R scripts to understand what the code is doing would be helpful.

Response:

We thank the reviewer for spotting this inconsistency, and we sincerely apologize for the oversight in providing the incorrect GitHub repository link. The correct link has been updated in the Data Availability Statement of the revised manuscript. The updated repository now includes clear documentation detailing the minimum requirements to run the code, an explanation of the code components, and instructions for reproducing the analysis presented in this study.

Reviewer #2 (Remarks to the Author):

While this manuscript presents original work and is worthy contribution to the literature, it is simply too long (>10000 words) in the current format.

The authors should make a choice of the aspects of their work they want to present, decrease the size of introduction and discussions. The results section should only present results and not reiterate the methods.

The manuscript has a very Belgian focus and doesn't discuss results in an international context. Have other research groups found similar results? Finally they did not discuss whether a similar analysis could be done using other data sources, e.g. from testing data or contact tracing data, which would confirm the findings that different age groups were more important during certain phases of the pandemic.

Response:

We acknowledge the reviewer's concern regarding the length and content of the manuscript. In response, we have substantially revised the manuscript, reducing the length of the Introduction, the Results, and Discussion sections. We have also restructured the paper by moving all methodological details to the dedicated Methods section, ensuring that the Results section focuses solely on the presentation of our findings. To aid in the understanding of the results, we have included a table (Table 1) that provides a clear guide to interpreting the key indices.

Regarding the focus of the manuscript, we appreciate the suggestion to place our findings in a broader international context. In the revised Discussion, we have explicitly compared our results with those of international studies that have used similar approaches to assess the impact of NPIs on population considering age groups. These comparisons can be found in lines 261-272 of the revised main text.

Reviewer #3 (Remarks to the Author):

This paper proposes a novel approach that focuses on the changing roles of different age groups in the transmission dynamics of SARS CoV-2 in Belgium from November 2020 to February 2022. The proposed method highlights the distinct roles played by various age groups in the virus spread.

Strengths:

- + The paper is well-written and easy to follow, and the problem setting and discussions are all clearly described.
- + Four different roles in transmission and survey waves for various age groups are well-presented.
- + The effectiveness of the proposed approach is supported by empirical results.
- + The contributions are significant and interesting for the epidemics and epidemiology domain.

Weaknesses:

- The proposed approach is based on the SEIR model. Ideally, if authors could demonstrate results and conclusions by using SIR, SEIS, and etc, that would be even better.

Response:

We appreciate the reviewer's feedback regarding the use of the SEIR model. However, we respectfully disagree with the suggestion that the SEIR framework is a weakness. On the contrary, the SEIR model we used is quite flexible and can be adapted to suit specific epidemiological contexts.

In our study, the SEIR model was deliberately chosen to accurately simulate the progression of susceptible individuals throughout the study period. The model, developed by Abrams et al. (2021) and extended by Willem et al. (2024), is specifically tailored to the COVID-19 context in Belgium and was deemed the most suitable for achieving the study's objectives. Simplifying it to an SIR or SEIS model could lead to misrepresentation of susceptibility dynamics across age groups, which is critical for the accurate assessment of age-specific contributions to transmission, as measured by the indices used in this study.

As discussed in the Methods section, under "Model Choices and Calibration," the structure of the NGM is primarily shaped by the infectious states of the model. Simplifying the model to an SIR or SEIS framework would still yield a NGM similar to the one presented in our study. However, the SEIR model offers additional accuracy by allowing a more detailed and realistic representation of the susceptible compartment.

Thus, while our methodology can be applied to various mathematical modeling frameworks, we remain confident that the SEIR model is the most appropriate choice for the specific context of this study.

-The proposed approach from the paper is not included in the comparison study in the evaluation.

Response:

We appreciate the reviewer's comment. While it is true that a formal comparative evaluation of our specific approach was not included in the original manuscript, we have made efforts to enrich both the Introduction and Discussion sections with references to studies that examine the role of various age groups using alternative methodologies. These references indicate that our findings are broadly consistent with those of other research, thereby offering valuable context to our analysis. We hope this expanded discussion meets the reviewer's expectations and provides a satisfactory response to this concern.

- The results need further explanation. For instance, why some mean R_t values are larger than 1 but some are less than 1 (for different waves) in Table 1?

Response:

We appreciate the reviewer's observation. In response, we have detailed the determination of R_t values in the Methods section, specifically in the paragraph "Model Choices and Calibration" (Lines 345-346). Additionally, we have updated the caption for Table 2 to emphasize that the R_t values are calibrated based on estimates derived from the number of positive PCR tests, as referenced in the cited study. We hope this clarification addresses the reviewer's concern.

- It would be helpful if the authors can provide the reasons why using Bayesian approach with MCMC sampling to calibrate the model.

Response:

We have detailed our rationale for using the Bayesian approach with MCMC sampling to calibrate the model in the Methods section, specifically in the paragraph "Model Choices and Calibration" (Lines 369-371).

References:

[2021] Abrams, Steven, et al. "Modelling the early phase of the Belgian COVID-19 epidemic using a stochastic compartmental model and studying its implied future trajectories." **Epidemics** 35 (2021): 100449. <https://doi.org/10.1016/j.epidem.2021.100449>

[2024] Willem, Lander, et al. "The impact of quality-adjusted life years on evaluating COVID-19 mitigation strategies: lessons from age-specific vaccination roll-out and variants of concern in Belgium (2020-2022)." **BMC public health** 24.1 (2024): 1171. <https://doi.org/10.1186/s12889-024-18576-w>

Response to Referees (December 2024):

"Insights into the role of children in the COVID-19 pandemic in Belgium: a longitudinal perturbation analysis"

Authors: Leonardo Angeli, Constantino Pereira Caetano, Nicolas Franco, Pietro Coletti, Christel Faes, Geert Molenberghs, Philippe Beutels, Steven Abrams, Lander Willem, Niel Hens

Dear Editor,

We sincerely thank you for the opportunity to revise our manuscript further. We also extend our gratitude to the reviewers for their constructive feedback and for recognising the improvements made in the revised version.

The reviewers' thoughtful comments have guided us in enhancing the clarity, consistency, and overall quality of our work. We conducted a second review of the manuscript to address all points raised, ensuring the alignment of values in the text and figures, providing additional clarification where needed, and improving the interpretation of key results.

Below, we provide a detailed, point-by-point response to each comment, outlining the specific changes made. We hope these revisions meet the expectations of the reviewers and the editorial board.

Kind regards,
The authors

Reviewer #1

Comment 1: Inconsistencies between values in text and figures

“There are many instances where values indicated in the main text don’t match what is reported in the figures (e.g., line 96, line 114, line 158). This makes it difficult to evaluate the consistency and interpretability of the results. Please ensure that these and any other inconsistencies are corrected.”

Response:

We thank the reviewer for highlighting these discrepancies. We sincerely apologize for the oversight in the initial revision and have conducted a comprehensive review of the manuscript to ensure consistency between the text, figures, and tables. Specifically, the inconsistencies identified in lines 96, 114, and 158 have been addressed, and the corresponding sections in the Results have been rephrased to reflect the corrections accurately.

All changes have been marked in blue (over the previous red text) in the revised manuscript (see *Marked_Manuscript.pdf*). These updates span lines 89 to 221 of the revised text.

Comment 2: Key contributors and interpretation of susceptibility in children

“The study concludes that children were “key contributors to SARS-CoV-2 transmission at various points throughout the observation period.” First, the term “key contributor” is broad, and it would be helpful to specify what this term refers to (line 193). Second, the interpretation of this finding is unclear regarding the fact that susceptibility remains high in children until the end of 2021. If they were key contributors to the epidemics, shouldn’t they be infected?”

Response:

We thank the reviewer for this insightful comment. To address the first point, we have revised the phrasing in the manuscript referring to children as “main contributors” for consistency. Specifically, we use the term to denote age groups exhibiting above-average sensitivity and elasticity indices at a given observation point, as defined at the beginning of the Results section. These correspond to groups where targeted interventions could lead to the largest variations in transmission potential at specific observation points. For the second point, we have expanded the discussion (lines 193–197 in the revised manuscript) to highlight the relative nature of the sensitivity and elasticity indices, at each observation point. These indices reflect the interplay between high contact intensity, waning immunity in other groups, and delayed vaccination efforts, which resulted in children playing a crucial role in transmission at specific points. Importantly, this contribution does not necessarily deplete the overall susceptibility of children, particularly given the dynamics of immunity waning and limited reinfections in this age group during the observation period. Moreover, the characterisation of children as main contributors should always be interpreted in the context of the transmission level at the corresponding time. For instance, children may be classified as main contributors during periods of lower R_t , which could lead to slower

depletion of susceptibles in this group.

Further elaboration on these dynamics and their implications for children's role in transmission is provided in the subsequent paragraphs of the Discussion (lines 198–232).

Comment 3: Role of SEIR model and PCR-based R_t estimates

“I still find certain paragraphs in the Methods section difficult to read. The role of the SEIR-like model versus R_t estimates from PCR data is unclear. Is the SEIR model only used to infer susceptibility over time? Is the R_t from PCR data only used to scale the NGM and avoid relying on R_t estimates from SEIR model simulations?”

Response:

We thank the reviewer for this observation. In the revised Methods section (Lines 376–387), we have clarified the distinct roles of the SEIR model and the PCR-based R_t estimates. The SEIR model is used to estimate age-specific susceptibility over time, integrating various data sources such as vaccination coverage, immunity waning, and variant-specific dynamics. This ensures a robust reconstruction of the evolving epidemiological landscape.

On the other hand, the PCR-based R_t estimates are employed to calibrate the dominant eigenvalue of the NGM. This ensures that the R_t values used in our analysis are consistent with observed epidemic trends, avoiding reliance on R_t estimates derived directly from SEIR model simulations. Furthermore, we note that the sensitivity indices derived in our study are invariant under the multiplication of the NGM (and R_t) by a nonzero scalar, ensuring the robustness of our analysis to this calibration approach.

Comment 4: Clarify the paragraph on school closures and transmission intensities

“Line 267-269: Please clarify the paragraph comparing the results to other studies. The phrasing is currently confusing regarding the relationship between school closures and changes in transmission intensities.”

Response:

We thank the reviewer for pointing out the need for clarification in this section. To avoid any potential misinterpretation, we have revised the paragraph (Lines 250–255) to focus on the observed effectiveness of school closures and social distancing in reducing transmission. We have opted to remove the part comparing the relative impacts of reopening schools and

broader social distancing measures, as the original phrasing relied on simulation-based findings that do not align with our study's scope.

Comment 5: Counterfactual analyses and missing legends

“I appreciate the new analyses with the counterfactual scenarios. However, some legends are missing in the corresponding figures (S3 to S6). It is unclear what the dotted colored lines represent. Please clarify.”

Response:

We thank the reviewer for bringing this to our attention and for their positive feedback on the counterfactual analyses. In response, we have updated the captions, legends, and annotations for Figures S3 to S6 in the Supplementary Material. The revised descriptions now clearly define all elements, including the dotted colored lines, axes, and their respective interpretations, to ensure clarity and improve the overall readability of these graphs.

Comment 6: Interpretation of elasticity gradient

I did not understand how the elasticity gradient should be interpreted. Please provide guidance on how to use this index and its interpretation.

Response:

We thank the reviewer for highlighting the need for further clarification on the elasticity gradient. In the revised Supplementary Methods (paragraph “Interpretation”), we have provided a detailed explanation of its significance and application. Specifically, the elasticity gradient offers insights into how variations in contact patterns and epidemiological parameters reshape the age-specific contribution to transmission, as captured by the cumulative elasticity indices at specific time points. This tool helps to explore the interplay between population groups and their relative roles in transmission hierarchies.

Additionally, we have adapted relevant parts of the Supplementary Discussion to ensure consistency and provide further context. All updates have been highlighted in blue (over the previous red text) in the updated *Marked_Sup_Mat.pdf* file.